# A strategy for tough and fatigue-resistant hydrogels via loose cross-linking and dense dehydration-induced entanglements

Danming Zhong [1,4], Zhicheng Wang [1,4], Junwei Xu[1], Junjie Liu[2], Rui Xiao[1], Shaoxing Qu [1,3] ✉ & Wei Yang [1]

Outstanding overall mechanical properties are essential for the successful utilization of hydrogels in advanced applications such as human-machine interfaces and soft robotics. However, conventional hydrogels suffer from fracture toughness-stiffness conflict and fatigue threshold-stiffness conflict, limiting their applicability. Simultaneously enhancing the fracture toughness, fatigue threshold, and stiffness of hydrogels, especially within a homogeneous single network structure, has proven to be a formidable challenge. In this work, we overcome this challenge through the design of a loosely cross-linked hydrogel with slight dehydration. Experimental results reveal that the slightly-dehydrated, loosely cross-linked polyacrylamide hydrogel, with an original/current water content of 87%/70%, exhibits improved mechanical properties, which is primarily attributed to the synergy between the long-chain structure and the dense dehydration-induced entanglements. Importantly, the creation of these microstructures does not require intricate design or processing. This simple approach holds significant potential for hydrogel applications where excellent anti-fracture and fatigue-resistant properties are necessary.

Hydrogels find applications in various fields such as human-machine interfaces[1–4], soft robotics[5–7], tissue engineering[8–10], and flexible electronics[11–13], where exceptional mechanical properties are essential. These properties include high fracture toughness, suitable modulus, high fatigue threshold, excellent stretchability, crack insensitivity, etc. However, conventional single network hydrogels tend to be either soft or brittle, resulting in conflicts between toughness and stiffness[14] as well as fatigue threshold and stiffness[15]. As indicated in Fig. 1a, the fracture toughness is generally divided into two parts: $\Gamma = \Gamma_0 + \Gamma_D$, where $\Gamma_0$ is the intrinsic fracture energy, i.e., the fatigue threshold of a hydrogel, and $\Gamma_D$ is the fracture energy contributing by the energy dissipation[16]. Most toughening strategies aim to incorporate energy dissipation mechanisms into hydrogels, while these methods often fail

to markedly improve the fatigue threshold[17]. Besides, the fracture toughness exhibits a negative correlation with the elastic modulus of a single network hydrogel[14], and the fatigue threshold is inversely linked to the square root of the modulus, as predicted by the Lake-Thomas model[18]. Achieving simultaneous improvements in these mechanical properties proves to be challenging. Taking the commonly employed polyacrylamide (PAAm) hydrogel as an example, the PAAm hydrogel with a medium degree of cross-linking and medium initial polymer content is referred to as the regular hydrogel (Fig. 1b). Eight additional hydrogels were prepared by adjusting both degrees of cross-linking and initial polymer contents. However, except for hydrogel 8, no other hydrogel demonstrates the simultaneous improvement in modulus and fracture toughness when compared to the regular hydrogel

[1]State Key Laboratory of Fluid Power & Mechatronic System, Key Laboratory of Soft Machines and Smart Devices of Zhejiang Province, Center for X-Mechanics, Department of Engineering Mechanics, Zhejiang University, Hangzhou 310027, China. [2]Applied Mechanics and Structure Safety Key Laboratory of Sichuan Province, School of Mechanics and Aerospace Engineering, Southwest Jiaotong University, Chengdu 611756, China. [3]Eye Center, The Second Affiliated Hospital, School of Medicine, Zhejiang University, Hangzhou 310009, China. [4]These authors contributed equally: Danming Zhong, Zhicheng Wang. ✉e-mail: squ@zju.edu.cn

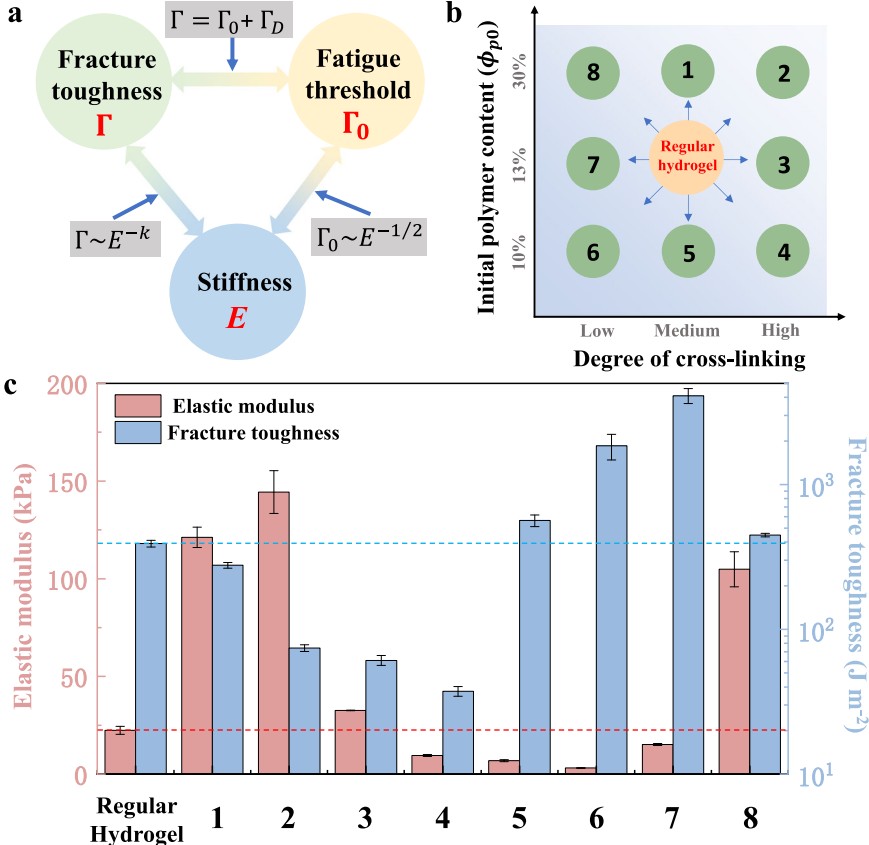

**Fig. 1 | Simultaneously improving the fracture toughness, fatigue threshold, and stiffness for a single network hydrogel is a challenge. a** The relationships between fracture toughness, fatigue threshold, and stiffness of hydrogels. **b** PAAm hydrogels with various degrees of cross-linking and initial polymer contents. The hydrogel with a medium degree of cross-linking and medium initial polymer content is referred to as the regular hydrogel. **c** Compared with regular hydrogel, adjusting the degree of cross-linking and initial polymer content cannot achieve significant improvement in fracture toughness and modulus simultaneously. Data are presented as mean values +/- SD ($n \geq 3$).

(Fig. 1c). And for hydrogel 8, the enhancement of fracture toughness is rather limited, amounting to less than 15%.

Recently, significant efforts have been made to enhance the stiffness, toughness, and fatigue resistance of hydrogels through the design of unconventional polymer networks[19,20]. One effective approach involves incorporating energy dissipation mechanisms into stretchy polymer networks, which significantly toughens hydrogels[16]. These dissipation mechanisms commonly involve the construction of unconventional networks and interactions, such as interpenetrating polymer networks[21,22], polymer networks with reversible cross-linkers[23,24], and polymer networks with high-functionality cross-linkers[25,26]. While these toughening mechanisms inhibit crack growth under monotonic loads, they frequently fail to resist crack growth under cyclic loads. Consequently, most of these toughening methods offer limited improvements in the fatigue threshold of hydrogels.

To improve both the fracture toughness and fatigue threshold of hydrogels, specific mechanisms have been designed at either microscopic or macroscopic levels, effectively immobilizing the fatigue crack. One approach involves constructing special microscopic structures such as nanocrystalline domains[27,28], separated microphases[29,30], and aligned nano/microfiber structures[31]. The energy required to damage these structures is significantly higher than that needed to fracture the corresponding amorphous polymer chains. For example, a PVA hydrogel with aligned nanofibers, prepared through freeze-thawing and mechanical training, reaches a fatigue threshold of 1250 J m$^{-2}$, which is orders of magnitude larger than that of a chemically cross-linked PVA hydrogel (~10 J m$^{-2}$)[31]. Another strategy involves designing macroscopic fiber/matrix composite structures[32,33], where both the fiber and matrix

are highly elastic and stretchable. The matrix is much softer and more stretchable than the fiber, and they are tightly bonded to create robust interfaces. When such a composite hydrogel is deformed, the soft matrix shears significantly, extending a large stretch over a long segment of the fiber at the crack tip[34]. Consequently, the stress concentration is alleviated, and the crack is immobilized by the fibers. By employing this strategy, a composite hydrogel with elastomer fibers achieves a fatigue threshold exceeding 1000 J m$^{-2}$ [35]. Recently, a strategy of crack tip softening for hydrogels has been proposed[17]. It enhances both the fracture toughness and fatigue threshold through elastic shielding and stress de-concentration at the crack tip.

Nevertheless, the above-mentioned strategies require the construction of heterogeneous structures, necessitating intricate design and complex production process. What's more, many of these strategies are limited to specific types of hydrogels. In contrast, a homogenous single network structure is much more desirable to process and applicable to most hydrogel systems. However, achieving simultaneous enhancements in fracture toughness, fatigue threshold, and stiffness using a homogeneous single network remains a formidable challenge due to the toughness/threshold-stiffness conflicts inherent in single-network hydrogels, as mentioned before.

Herein we propose a simple strategy. A hydrogel is polymerized with a low degree of cross-linking and then slightly dehydrated. We ascertain that dehydration creates dense entanglements that stiffen a hydrogel. Additionally, these entanglements effectively dissipate energy when large deformation is applied, resulting in high fracture toughness. Moreover, long polymer chains ensure a high fatigue threshold, and the increase in chain density by dehydration further

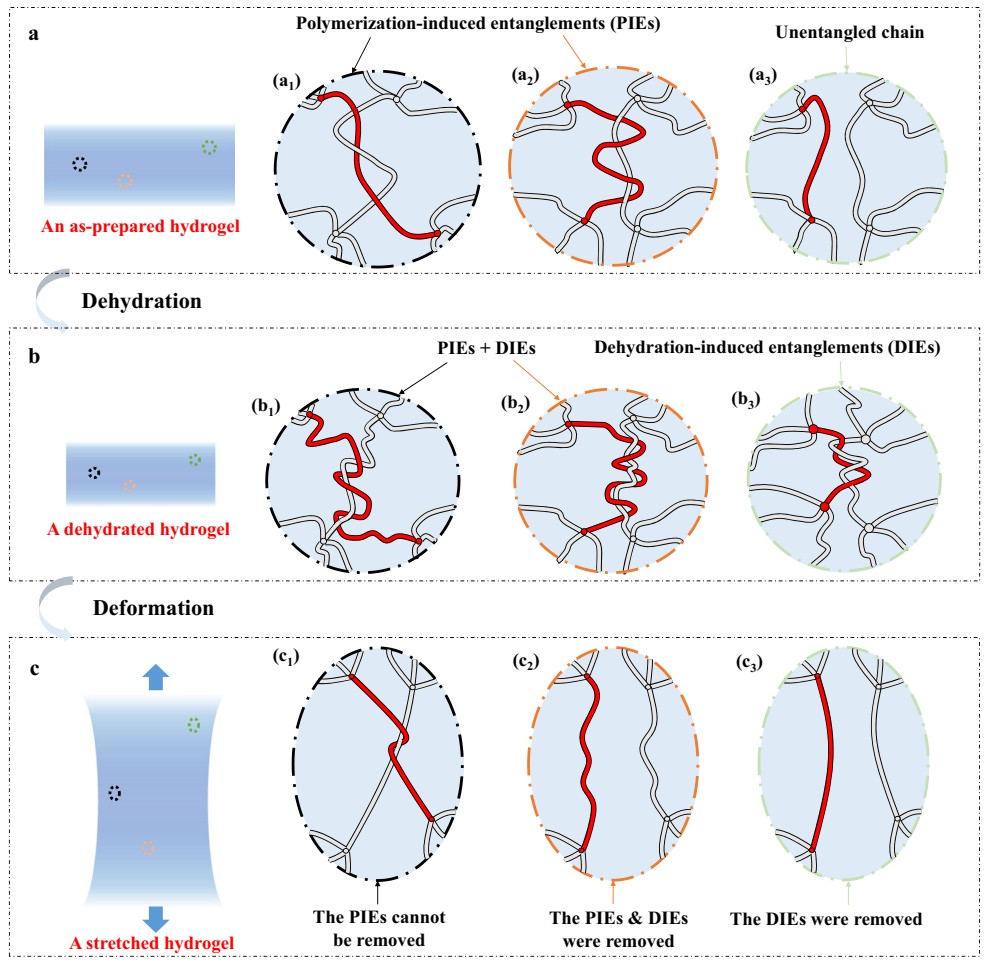

**Fig. 2 | Schematic diagram of entanglements in a hydrogel. a** Polymerization-induced entanglements (PIEs) are formed in an as-prepared hydrogel: (a1) Some PIEs cannot be removed unless chain scission happens. (a2) Some PIEs can be removed by external forces. (a3) Some chains are not entangled. **b** Dehydration-induced entanglements (DIEs) are introduced during hydrogel dehydration: (b1-b2) Additional DIEs are introduced based on the existing PIEs. (b3) Some initially unentangled chains form DIEs. **c** The dehydrated hydrogel is stretched: (c1) The DIEs are disentangled but the PIEs are not removed. (c2) Both PIEs and DIEs are removed by external forces. (c3) The DIEs are removed by external forces.

improves the threshold. We demonstrate this strategy by taking PAAm hydrogel as a model material. A loosely cross-linked hydrogel with an initial water content ($\phi_{w0}$) of 87% was dehydrated to a current water content ($\phi_w$) of 70%. The resulting hydrogel possesses a modulus of ~90 kPa, a fracture toughness of ~22000 J m$^{-2}$, and a fatigue threshold of ~300 J m$^{-2}$. It also displays a fracture stretch larger than 20 and exhibits crack-insensitivity. Adopting this strategy, a conventional single network hydrogel achieves an ultra-high toughness and a high fatigue threshold, while breaking the toughness-stiffness conflict and the fatigue threshold-stiffness conflict which can hardly be resolved by a single network hydrogel.

## Results

### Dehydration-induced entanglements (DIEs)
Our strategy comprises two essential elements. The first key point involves maintaining a low degree of cross-linking, which results in the formation of a long-chain structure. During the polymerization of a hydrogel, monomers connect into chains, and those chains are cross-linked into a three-dimensional network. Simultaneously, interchain entanglements are formed (Fig. 2a). It is worth noting that long chains are more susceptible to entanglement compared to short chains. The entanglements formed during the polymerization process are labeled as polymerization-induced entanglements (PIEs). Certain PIEs cannot be removed, unless the chains are broken (Fig. 2a1). These inextricable

entanglements are common in hydrogels prepared with a high polymer content[14]. Nevertheless, some PIEs can be removed through the application of external forces (Fig. 2a2). And there are also chains that remain unentangled (Fig. 2a3). The polymerization-induced entanglements are well understood with the development of polymer physics[36–38].

The second key point in our strategy centers on the dehydration-induced entanglements (DIEs), which, despite some recent research[39,40], have not received full appreciation. When dehydrated, a hydrogel shrinks (Fig. 2b). The end-to-end distance of each single chain decreases, causing the chain to adopt a more curly conformation. In the meantime, the polymer chains get closer to each other. As a result, for the chains with PIEs already, additional DIEs are introduced (Fig. 2b1 and b2). And some originally unentangled chains form DIEs (Fig. 2b3)[41]. When the dehydrated hydrogel is stretched, all the DIEs are removable, but some PIEs cannot be removed (Fig. 2c). The PIEs demonstrated by Fig. 2a1 are not removed, and they work as topological cross-linkers (Fig. 2c1). Other PIEs and DIEs are disentangled by external force (Fig. 2c2 and c3).

### Entanglements-related hyperelasticity and stiffening
We first demonstrate the influence of the degree of cross-linking and dehydration on hyperelasticity and modulus of hydrogels. We prepared hydrogels with three different degrees of cross-linking, and the

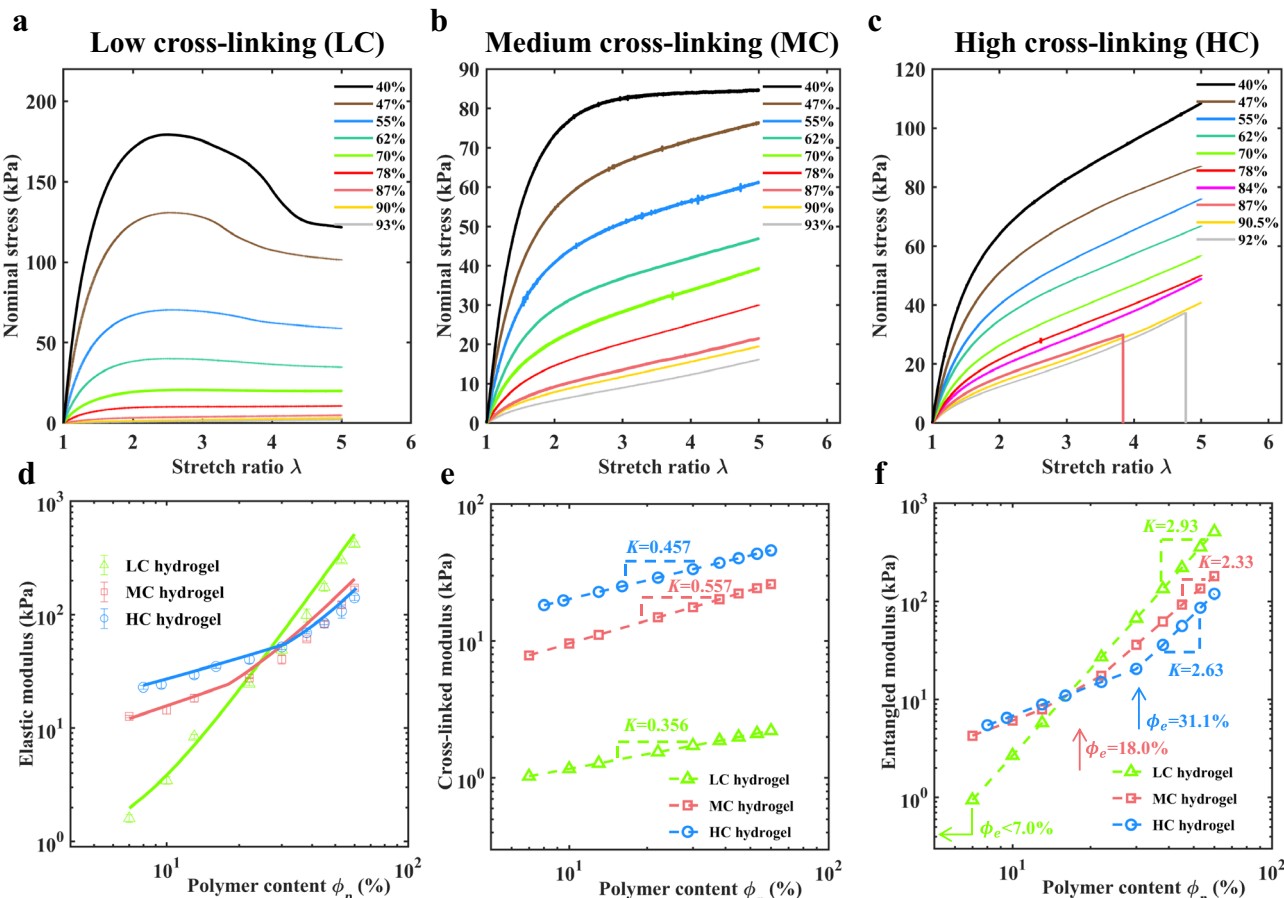

**Fig. 3 | Quasi-static nominal stress-stretch ratio curves and moduli.** Uniaxially quasi-static nominal stress-stretch ratio curves (s-s curves) for **a** Low cross-linked (LC) hydrogels, **b** Medium cross-linked (MC) hydrogels, and **c** High cross-linked (HC) hydrogels with various water contents $\phi_w$. **d** Experimental results and theoretical predictions of elastic modulus ($E$) - polymer content ($\phi_p$) relationships for

LC/MC/HC hydrogels. Experimental data are presented as mean values +/- SD ($n \geq 3$). **e** Theoretical cross-linked modulus ($E_c$) - polymer content ($\phi_p$) relationships for LC/MC/HC hydrogels. **f** Theoretical entangled modulus ($E_e$) - polymer content ($\phi_p$) relationships for LC/MC/HC hydrogels.

initial water content was 87%. They are named high cross-linked (HC) hydrogels, medium cross-linked (MC) hydrogels, and low (loosely) cross-linked (LC) hydrogels, respectively. The as-prepared MC hydrogel with $\phi_{w0} = 87\%$ is just the regular hydrogel mentioned in Fig. 1b. The as-prepared hydrogel samples were dehydrated/swelled to lower/ higher water contents, and then stretched uniaxially by quasi-static loading. The corresponding quasi-static nominal stress-stretch ratio curves (s-s curves) for LC/MC/HC hydrogels with various water contents ($\phi_w$) are exhibited in Fig. 3a-c, respectively.

For the LC hydrogels, as shown in Fig. 3a, the quasi-static s-s curve increases significantly when the hydrogel is dehydrated from $\phi_{w0} = 87\%$ to $\phi_w = 40\%$. Furthermore, the phenomenon of strain-softening, characterized by a decrease in tangent modulus with an increasing stretch ratio, becomes more pronounced as dehydration increases. This strain-softening behavior is closely associated with interchain entanglements[42–44]. As the external load increases, the DIEs are removed gradually, leading to the continuous decrease of the tangent modulus of a s-s curve[41]. For certain LC hydrogels with relatively low $\phi_w$, the stress even drops when the stretch ratio increases. A lower $\phi_w$ corresponds to more DIEs, and their disentanglement under external loading results in a more pronounced strain-softening behavior. As a comparison, the MC/HC hydrogels show less strain-softening (Fig. 3b and c) when dehydrated to the same $\phi_w$ as the LC hydrogels, mainly due to the fact that the shorter chains are harder to form interchain entanglements. Further analyses will be given below.

The DIEs contribute to the elastic modulus. A low degree of cross-linking is more conducive to stiffening the hydrogel by dehydration. Figure 3d shows the elastic modulus ($E$)-polymer content ($\phi_p$) relationships for LC/MC/HC hydrogels in a log-log plot. The data points represent the experimental results, and the solid lines depict the theoretical predictions based on a constitutive model[41]. It is worth noting that the moduli shown in Fig. 3 were derived from the quasi-static s-s curves. Therefore, the moduli presented here do not include the contribution of viscoelasticity. The log($E$)-log($\phi_p$) curve of LC hydrogels is much steeper than those of MC/HC hydrogels. For as-prepared hydrogels ($\phi_{w0} = 87\%$, i.e., $\phi_{p0} = 13\%$), the difference between their moduli is evident, and the higher the degree of cross-linking, the larger the modulus. When dehydrated to a water content of $\phi_w = 70\%$, these hydrogels exhibit similar moduli. However, with further dehydration, we found that the lower the degree of cross-linking, the larger the modulus. The modulus of a LC hydrogel increases rapidly as the water content decreases, which is beneficial to achieve stiffening by dehydration. For instance, the modulus of a LC hydrogel increases five folds, from 8.4 kPa to 50.9 kPa, when the water content drops from 87% to 70%. In contrast, the modulus of a HC hydrogel only increases by 0.76 times after the same dehydration process.

The variation of $E$-$\phi_p$ curves with various cross-linking degrees shown in Fig. 3d was further analyzed in conjunction with a constitutive model. Theoretically, the modulus of a hydrogel consists of the contribution from the cross-linked networks (the cross-linked modulus, $E_c$) and the contribution from the interchain entanglements

(the entangled modulus, $E_e$). When dehydrated, both $E_c$ and $E_e$ increase, satisfying corresponding scaling laws[41]. According to the polymer physics[36], there is a critical polymer content, $\phi_e$, which corresponds to the boundary between unentangled and entangled regimes. The modulus of a hydrogel is dominated by $E_c$ in the unentangled regime ($\phi_p < \phi_e$), while dominated by $E_e$ in the entangled regime ($\phi_p > \phi_e$)[38]. When $\phi_p > \phi_e$, the entangled modulus increases greatly with dehydration. The scaling laws between $E_c$, $E_e$, and $\phi_p$ are embedded into a hyperelasticity model (More details can be found in Supplementary Note 1).

We compare this model with the quasi-static s-s curves presented in Fig. 3a-c (See Supplementary Fig. 1 and Supplementary Fig. 2). The fitted model parameters (See Supplementary Table 1) are adopted to predict the theoretical $E$-$\phi_p$ curves (Fig. 3d), $E_c$-$\phi_p$ curves (Fig. 3e), and $E_e$-$\phi_p$ curves (Fig. 3f). According to Fig. 3e, the log($E_c$)-log($\phi_p$) curves for hydrogels with different degrees of cross-linking appear linear, exhibiting little difference in slope. By comparing Fig. 3d and Fig. 3f, we observed that the evolution of elastic modulus with $\phi_p$ is mainly dominated by the evolution law between $E_e$ and $\phi_p$. The slopes of the log($E_e$)-log($\phi_p$) curves for LC/MC/HC hydrogels do not differ significantly when $\phi_p > \phi_e$ (Fig. 3f). However, the critical polymer content of these hydrogels varies greatly. The HC hydrogels and MC hydrogels possess critical polymer contents of $\phi_e = 31.1\%$ and $\phi_e = 18.0\%$, respectively, while the $\phi_e$ of the LC hydrogels is expected to be less than 7% (See Supplementary Note 2 and Supplementary Fig. 1 for more detailed analysis). Accordingly, hydrogels with shorter chains are more difficult to form abundant entanglements by dehydration. Even when a HC hydrogel is dehydrated to $\phi_w = 70\%$, substantial entanglements are not observed (the ratio of entangled modulus to cross-linked modulus is $\alpha_{en} = 0.61$, indicating that the entangled modulus is smaller than the cross-linked modulus) due to the current polymer content ($\phi_p = 30\%$) still being below the critical polymer content ($\phi_e = 31.1\%$). On the contrary, quite a few entanglements exist in the as-prepared LC hydrogel, given that the critical polymer content ($\phi_e < 7\%$) is smaller than the initial polymer content ($\phi_{p0} = 13\%$). As a result, once dehydrated, the DIEs arise instantly, leading to a rapid increase in the entangled modulus.

In addition, the contribution of entanglements to the elastic modulus rises vastly with progressive dehydration. As depicted in Fig. 3e and f, for LC/MC/HC hydrogels, The slopes of the log($E_e$)-log($\phi_p$) curves ($k = 2.93$, $2.33$, and $2.63$, respectively) are much larger than those of the log($E_c$)-log($\phi_p$) curves ($k = 0.36$, $0.56$, and $0.46$, respectively) when $\phi_p > \phi_e$. For example, in the case of a LC hydrogel, the ratio of entangled modulus to cross-linked modulus ($\alpha_{en}$) escalates fast from 4.5 to 38.9 as the water content reduces to 70%. Without the presence of substantial entanglements, the enhancement of stiffness with dehydration would be much slower.

In conclusion, the stiffening of LC hydrogels by slight dehydration is quite efficient, thanks to the low critical polymer content of the long-chain structure and the sharp enhancement of entangled modulus by dehydration.

### Fracture toughness, crack-insensitivity, and energy dissipation

A hydrogel with a conventional single network structure encounters stiffness ($E$)-fracture stretch ($\lambda_f$) conflict. Consider a hydrogel with $n$ polymer chains per unit volume and $N$ monomers on each chain, the elastic modulus $E$ is inversely proportional to the chain length $N$, while the fracture stretch $\lambda_f$ is proportional to $\sqrt{N}$. If the chain length $N$ decreases, the modulus increases, but the fracture stretch decreases. This $E$-$\lambda_f$ conflict is clearly demonstrated by the s-s curves of the as-prepared LC/MC/HC hydrogels in Fig. 4a. Taking the current strategy, a LC hydrogel was mildly dehydrated to $\phi_w = 70\%$, the resulting hydrogel resolves the $E$-$\lambda_f$ conflict. As shown in Fig. 4a, the s-s curves of the dehydrated LC hydrogel (black lines) completely encircle those of the as-prepared LC hydrogel (blue lines), indicating that the modulus,

stretchability, strength, and work of fracture are simultaneously improved by mild dehydration. Besides, the slightly-dehydrated LC hydrogel (black lines) is stiffer than the as-prepared HC hydrogel (red lines), even though the density of the cross-linkers of the latter is 14 times that of the former.

We measured the fracture toughness of LC/MC/HC hydrogels with various water contents by pure shear tests. A sample with a width of 50 mm, a height of 10 mm, and a thickness of about 2 mm was stretched along the height direction, and the sample fractured at a stretch ratio of $\lambda_f$. Another specimen, with identical dimensions, was prefabricated with a 20 mm-long crack. This precut sample was stretched along the height direction, and the fracture stretch was labeled as $\lambda_c$. Then the fracture toughness values were calculated (See Methods and Supplementary Figs. 3–5) and presented in Fig. 4b. The results reveal that the fracture toughnesses of MC/HC hydrogels are greatly improved with dehydration. The toughness of the MC/HC hydrogels increases 5.9 times and 24.8 times, respectively, when water content reduces from $\phi_{w0} = 87\%$ to $\phi_w = 40\%$. However, the toughness of these hydrogels remains below 3000 J m$^{-2}$. The LC hydrogels are also toughened when slightly dehydrated. A LC hydrogel with $\phi_w = 70\%$ reaches a fracture toughness of 9782.5 J m$^{-2}$. Such a high fracture toughness arises from the synergy of multiple mechanisms as will be enumerated below.

First, this hydrogel is highly stretchable. As mentioned above, LC hydrogels possess longer polymer chains, leading to a larger fracture stretch. As evident in Fig. 4c, the fracture stretch of the as-prepared LC hydrogel is $\lambda_f = 17.9$, while the slightly-dehydrated hydrogel ($\phi_w = 70\%$) is even more stretchable, with $\lambda_f = 20.6$. The shrinkage of the hydrogel causes the polymer chains to curl up further, demanding additional deformation to reach their stretch limit. Consequently, the dehydrated hydrogel becomes more stretchable, contributing to its improved toughness.

Second, a mildly-dehydrated LC hydrogel is quite crack-insensitive. As illustrated in Fig. 4d, a slightly-dehydrated LC hydrogel ($\phi_w = 70\%$) was prefabricated with a 20 mm-long crack. As the stretch increases, the crack gradually blunts without propagating, even if the stretch ratio reaches 20. The sample finally fails close to the clamp. Typically, a large crack greatly reduces the stretchability of a hydrogel, i.e., $\lambda_c \ll \lambda_f$ (As seen in the fracture stretches of MC/HC hydrogels in Fig. 4c). However, the fracture stretch of a precut LC hydrogel with $\phi_w = 70\%$ is very close to that of an uncut sample (with $\lambda_c = 20.2$ and $\lambda_f = 20.6$, respectively. See Fig. 4c). And the precut LC hydrogel ($\phi_{w0} = 87\%$) can even stretch slightly further than the corresponding uncut hydrogel (with $\lambda_c = 19.2$ and $\lambda_f = 17.9$, respectively. See Fig. 4c), indicating a strong crack-insensitivity. The proximity between $\lambda_c$ and $\lambda_f$ for LC hydrogels with high water content is attributed to their large fractocohesive length. The fractocohesive length, $R_f$, defined as the ratio of the fracture toughness to the work of fracture, describes a material-specific length that marks the transition from flaw-insensitive to flaw-sensitive fracture[45]. To verify, larger samples of LC hydrogels with $\phi_w = 70\%$ (a width of 120 mm, a height of 40 mm, a thickness of about 3 mm, and a 48 mm-long crack for precut samples) were tested. The results reveal smaller fracture stretches for precut hydrogels (with $\lambda_c = 17.8$ and $\lambda_f = 22.1$, respectively. See Supplementary Fig. 6). The precut samples fractured due to the propagation of the crack. The calculated fractocohesive length of LC hydrogels ($\phi_w = 70\%$) is determined to be 28.5 mm. This value exceeds the crack length of the small samples (20 mm) used in pure shear tests, aligning with the observed crack-insensitivity of small samples.

The third mechanism underlying the ultra-large toughness is the combination of a large fracture process zone and a strong energy dissipation capacity. When the material surrounding the crack tip undergoes large deformation, the dehydration-induced entanglements are gradually removed. This disentanglement process, which involves the movement and friction of polymer chains, consumes a

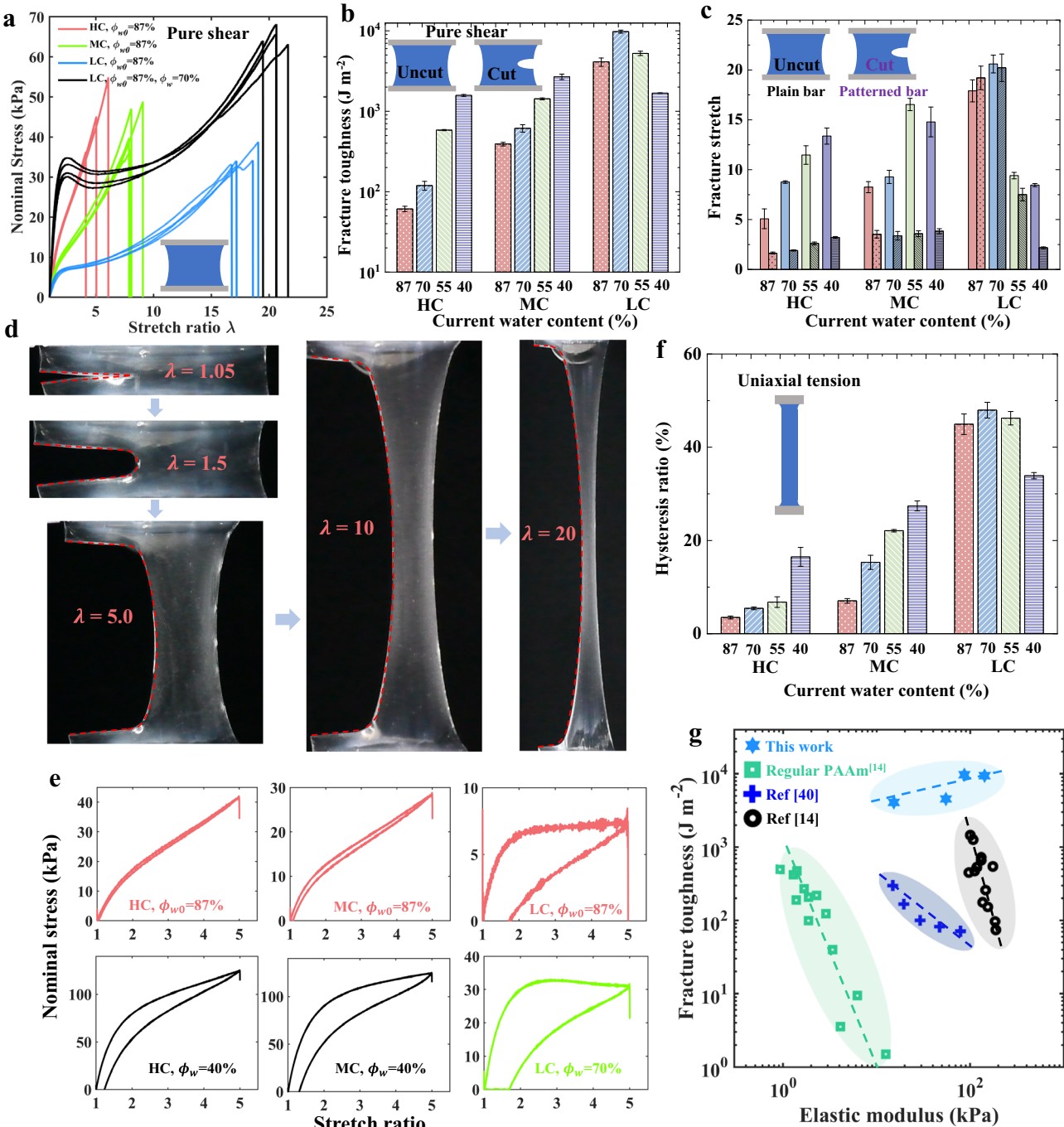

**Fig. 4 | Fracture and energy dissipation of hydrogels. a** The s-s curves of the as-prepared LC/MC/HC hydrogels and slightly-dehydrated LC hydrogels ($\phi_w = 70\%$) by pure shear tests. The slightly-dehydrated LC hydrogel exhibits larger modulus, fracture stretch, strength, and work of fracture. **b** The fracture toughness of LC/MC/HC hydrogels with various $\phi_w$ values measured by pure shear tests. Data are presented as mean values +/− SD ($n \geq 3$). **c** The fracture stretches for uncut samples (the plain bars) and precut samples (the patterned bars) of LC/MC/HC hydrogels with various $\phi_w$ values. Data are presented as mean values +/− SD ($n \geq 3$). **d** The snapshots of a slightly-dehydrated LC hydrogel ($\phi_w = 70\%$) under various

deformation. The crack does not grow at a stretch ratio of 20. **e** The s-s curve in a loading-unloading cycle at a strain rate of $\dot{\varepsilon} = 0.1\,\mathrm{s}^{-1}$. The LC hydrogels show larger hysteresis loops. **f** The hysteresis ratios of LC/MC/HC hydrogels with various $\phi_w$ values measured by uniaxial tension tests. Data are presented as mean values +/− SD ($n \geq 3$). **g** Four families of PAAm hydrogels are plotted on the toughness-modulus plane. The slightly-dehydrated LC hydrogels by the current strategy, with $\phi_w$ ranging from 65% to 87%, break the toughness-stiffness conflict, while the other strategies did not.

substantial amount of energy. It should be noted that the fractocohesive length also characterizes the size of the fracture process zone (FPZ). Though only a small fraction of chains in the FPZ break, all the energy dissipated in the FPZ contributes to the fracture toughness[46]. The fractocohesive length of a mildly-dehydrated LC hydrogel (28.5 mm) is much larger than that of MC/HC hydrogels (ranging from

0.3 to 2.5 mm. See Supplementary Fig. 7). Moreover, LC hydrogels possess a better capacity to dissipate energy. As shown in Fig. 4e, the as-prepared MC/HC hydrogels are highly elastic, with tiny hysteresis loops between loading and unloading curves. When they are dehydrated to a low water content of 40%, medium hysteresis loops are observed. In contrast, both the as-prepared LC hydrogel and the

slightly-dehydrated LC hydrogel exhibit large hysteresis loops. The hysteresis ratio, defined as the area ratio between the hysteresis loop and the region under the loading curve, is listed in Fig. 4f. The slightly-dehydrated LC hydrogel ($\phi_w = 70\%$) demonstrates optimal energy dissipation capacity, with the dissipated energy accounting for approximately half of the work performed by the external force.

A large FPZ around the crack tip dissipates energy substantially, so that the stress concentration at the crack tip is effectively alleviated, and the hydrogel is greatly toughened. The fractocohesive length of LC hydrogels with $\phi_w = 70\%$ ($R_f = 28.5$ mm) lies between the height of small samples ($H = 10.0$ mm) and the height of large samples ($H = 40$ mm). The size of the FPZ for large samples ($R_f = 28.5$ mm) is much larger than that of the small samples ($R_f = 10.0$ mm). With more materials to dissipate energy, the measured toughness of the large samples is much higher, being $\Gamma = 21752$ J m$^{-2}$ (See Supplementary Fig. 6). We also tested the toughness of the as-prepared LC hydrogels using large samples (a width of 150 mm, a height of 40 mm, a thickness of about 3 mm, and a 60 mm-long crack for precut samples), the measured toughness is $\Gamma = 10152$ J m$^{-2}$ (See Supplementary Fig. 8), which is about 2.5 times larger than the toughness of small samples ($\Gamma = 4109.6$ J m$^{-2}$). Obviously, the fracture toughness measured by pure shear tests is dependent on the sample size. Similar phenomena were observed in the measurement of toughness by the 90-degree peeling test[46] and 180-degree peeling test[47]. The fractocohesive length is a critical size; below this size, the toughness increases with the sample size, while above it, the toughness reaches a plateau. We emphasize that in the comparison of fracture toughness values for different hydrogels tested in this paper, a fair comparison is ensured by considering the measured fracture toughness values for samples of identical dimensions ($50 \times 10 \times 2$ mm). For example, the fracture toughness of the LC hydrogel with $\phi_w = 70\%$ is reported as $\Gamma = 9782.5$ J m$^{-2}$ rather than $\Gamma = 21752$ J m$^{-2}$ in Fig. 4b and g.

The slightly-dehydrated LC hydrogels, with a relatively high content of water, resolve the stiffness-fracture toughness conflict. The LC hydrogels were prepared and then dehydrated to different water content: $\phi_w = 75\%$, 70%, and 65%, respectively. The moduli and fracture toughnesses of these hydrogels, along with some other PAAm hydrogels from references[14,40], are listed in Fig. 4g. Regular PAAm hydrogels are either soft or brittle. Certain PAAm hydrogels with specific designs and processing have achieved higher toughness and modulus than regular PAAm hydrogels[14,40]. In contrast, adopting the current strategy, the fracture toughness of the LC hydrogel with $\phi_w = 70\%$ approaches ten thousand J m$^{-2}$. The LC hydrogel with $\phi_w = 65\%$ reaches a high modulus (143.1 kPa) and a large toughness (9489.6 J m$^{-2}$) simultaneously. The dense entanglements induced by dehydration not only stiffen but also toughen the hydrogel, breaking the stiffness-toughness conflict.

## Fatigue-resistance

We tested the fatigue threshold $\Gamma_0$ of the slightly-dehydrated LC hydrogel ($\phi_w = 70\%$). As illustrated in Fig. 5a, precut samples were cyclically loaded with various maximal stretch ratios $\lambda_m$, and the crack growth processes were recorded. To calculate the corresponding energy release rates $G$, another uncut sample was also stretched cyclically. After 100 cycles with a maximal stretch ratio of $\lambda_x$, we upscaled the maximal stretch ratio to a larger value (Fig. 5b). The values of the maximal stretch ratio adopted here correspond to the values of $\lambda_m$ in fatigue tests. During the initial 10 cycles, the maximal stress in each cycle obviously attenuated, eventually stabilizing after 100 cycles (See Fig. 5c and Supplementary Fig. 9). In the 100$^{th}$ cycle of the cyclic experiments with a maximal stretch ratio of $\lambda_m = 2.65$, a stable hysteresis loop was observed (Fig. 5d). A similar hysteresis loop was also observed in the 100th cycle of the cyclic experiments with $\lambda_m = 4.70$. We believe that some entanglements are progressively removed with each cycle, leading to the gradual attenuation of the

maximal stress (Fig. 5c). Some entanglements are reversible, as they are removed during loading and reformed during unloading, resulting in the stable hysteresis loop (Fig. 5d).

A precut LC hydrogel with $\phi_w = 70\%$ was cyclically loaded with $\lambda_m = 2.65$ (i.e., an energy release rate of 256.1 J m$^{-2}$). As shown in Fig. 5e, after five thousand cycles, the crack propagated by about 1 mm. However, upon comparing snapshots taken in the 5000$^{th}$ and 20000$^{th}$ cycles, it was evident that the crack was then immobilized. When a larger $\lambda_m$ was applied ($\lambda_m = 3.90$, i.e., energy release rate $G = 425.3$ J m$^{-2}$), the crack grew fast during the initial few thousand cycles, followed by steady growth thereafter (Fig. 5f). The steady growth region was adopted to calculate the crack growth velocity. Figure 5g plots the relation between the crack extension per cycle (dc/dN) and the energy release rate $G$. The fatigue threshold was determined to be 291.4 J m$^{-2}$, a large value considering that conventional PAAm hydrogels and natural rubber typically have fatigue thresholds of around ~10 J m$^{-2}$ and ~50 J m$^{-2}$, respectively.

The low degree of cross-linking facilitates long polymer chains, which endow the LC hydrogels with a high fatigue threshold. A fatigue crack grows by breaking only a single layer of deformed chains across the crack plane. According to the Lake-Thomas model[18], the fatigue threshold is equivalent to the energy stored in a layer of polymer chains per unit area[19,48]: $\Gamma_0 = \phi_p^{2/3} l\sqrt{N}J/V$, where $\phi_p$ represents the polymer content of a hydrogel, $l$ is the length of a monomer, $N$ is the number of monomers in a single chain, $J$ is the chemical bond energy, and $V$ is the volume of a monomer. Obviously, the longer the polymer chains (i.e., larger $N$), the larger the $\Gamma_0$. Thus, a long-chain structure serves as the basis for fatigue-resistant hydrogels. However, simply increasing the chain length does not lead to a significant improvement in the fatigue threshold. In our work, the chain length of the as-prepared LC hydrogels is 30 times that of the as-prepared HC hydrogels, which means the enhancement of the threshold is less than 6 times ($\Gamma_0 \sim \sqrt{N}$). Since the fatigue crack breaks a layer of chains across the crack plane, increasing the chain density (i.e., the polymer content) can raise the threshold. Dehydration of a hydrogel enlarges the chain density, thereby further contributing to the fatigue threshold. Studies have shown that the classical Lake-Thomas model agrees well with measured $\Gamma_0$ of hydrogels with high water content, but it underestimates the $\Gamma_0$ when the water content decreases[48]. Recent research demonstrated that the threshold of a dehydrated hydrogel follows the scaling law[49]: $\Gamma_0(\phi_p) \sim \phi_p^{3\nu/(3\nu-1)}$, where $\nu$ is the same scaling exponent as adopted in the $G_c$-$\phi_p$ and $G_e$-$\phi_p$ relationships (See Eqs. 4 and 5 in Supplementary Note 1). Interestingly, the scaling law for $\Gamma_0$ is similar to the scaling law for $G_e$ when $\phi_p > \phi_e$ (Eq. 5 in Supplementary Note 1). As discussed above, the modified scaling law between the threshold and chain length and current polymer content is as follows: $\Gamma_0(\phi_p) \sim \sqrt{N}\phi_p^{3\nu/(3\nu-1)}$. The scaling exponent for LC hydrogels is $\nu = 0.506$ (Supplementary Table 1). According to the modified scaling law, compared with the regular hydrogel, the fatigue threshold of the dehydrated LC hydrogel ($\phi_w = 70\%$) can be increased by 36.7 times, which is comparable with the observed 27-fold improvement by experiment. It is evident that the combination of a long-chain structure and dehydration-induced increment of chain density significantly improves the fatigue threshold.

## Overall mechanical properties

The current strategy fabricated hydrogels with excellent overall mechanical properties. The mechanical properties of four PAAm hydrogels were compared: hydrogel $A$ with a water content of $\phi_{w0} = 70\%$, hydrogel $B$ with a water content of $\phi_{w0} = 87\%$, hydrogel $C$ prepared by the current strategy, with an initial water content of $\phi_{w0} = 87\%$ and a current water content of $\phi_w = 70\%$, and a regular hydrogel with medium cross-linking and $\phi_{w0} = 87\%$ (i.e., the regular hydrogel shown in Fig. 1b). Hydrogel $A$, $B$, and $C$ were prepared with low cross-linking (Supplementary Fig. 10a). Hydrogel $C$ was

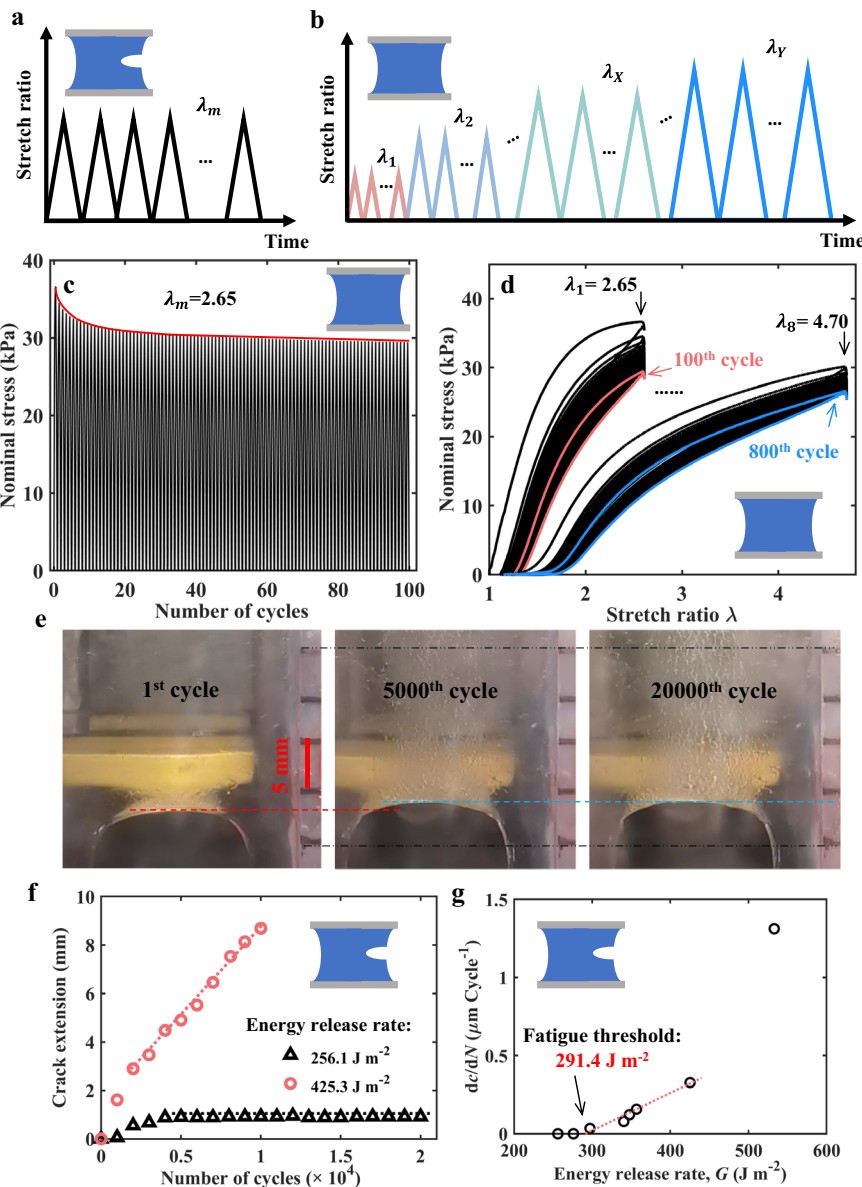

**Fig. 5 | Fatigue-resistance of slightly-dehydrated LC hydrogels. a** A precut slightly-dehydrated LC hydrogel ($\phi_w$ = 70%) is cyclically loaded with a maximal stretch ratio of $\lambda_m$, and the crack growth process is recorded. **b** An uncut sample is cyclically loaded with an increasing maximal stretch ratio to measure the corresponding energy release rates. **c** With $\lambda_m$ = 2.65, the maximal stress of each cycle decays to a stable value after 100 cycles. **d** Stable hysteresis loops are observed after 100 cycles with certain $\lambda_m$. **e** A precut sample is loaded with an energy release rate of 256.1 J m$^{-2}$. The crack grows during the initial few thousand cycles and then is immobilized. **f** The crack growth process of precut samples under various energy release rates. **g** The relationship between crack extension per cycle (d$c$/d$N$) and the energy release rate ($G$). The calculated fatigue threshold is 291.4 J m$^{-2}$.

dehydrated from hydrogel *B*. By slight dehydration, the overall mechanical properties, including the modulus, fracture stretch, strength, work of fracture, fracture toughness, and fatigue threshold, are simultaneously improved (See Fig. 4a and Supplementary Table 2). In comparison with the regular hydrogel, the fracture toughness, fatigue threshold, and modulus of hydrogel *C* increased by 25.0 times, 27.0 times, and 3.9 times, respectively. Obviously, the current strategy breaks the toughness-stiffness and fatigue threshold-stiffness conflicts. Moreover, the enhancement of the fatigue threshold is comparable with the increase in toughness. From another perspective, hydrogel *A* and hydrogel *C* share identical chemical components, i.e., the current water content of $\phi_w$ = 70%, a cross-linker-to-monomer molar ratio of $5.0 \times 10^{-5}$, and an initiator-to-monomer molar ratio of $1.0 \times 10^{-3}$. However, the processing and microstructure are quite different. Owing to the lower initial polymer content of hydrogel *C*, its

modulus is slightly smaller than that of hydrogel *A*, measuring at 83% of the modulus of hydrogel *A*, and this aligns with the theory of polymer physics[36]. Nevertheless, hydrogel *C* is much tougher, with a fracture toughness of 21.8 times that of hydrogel *A*. In addition, its fatigue threshold is 4.1 times that of hydrogel *A*.

As a demonstration, these four hydrogels were fabricated to lift a weight. Each hydrogel sample, with a width of 50 mm, a height of 10 mm, and a thickness of about 2 mm, was precut with a 20 mm-long crack and then stretched to lift a weight. The force-displacement curves are depicted in Supplementary Fig. 10b. In Supplementary Movie 1, it is evident that the crack does not propagate in hydrogel *B* due to its large toughness. However, hydrogel *B* fails to lift a 300 g weight as it is soft and weak, ultimately fracturing near the fixture (Supplementary Fig. 10c). Supplementary Movie 2 reveals that hydrogel *A* is also unable to lift a 300 g weight. Despite its higher modulus

compared to hydrogel *B*, hydrogel *A* is too brittle, resulting in crack propagation (Supplementary Fig. 10d). The regular hydrogel performs even worse (see Supplementary Movie 3 and Supplementary Fig. 10b). By contrast, hydrogel *C* successfully lifts a 500 g weight (see Supplementary Movie 4 and Supplementary Fig. 10e), indicating that the hydrogel designed by the current strategy is both tough and strong.

## Discussion

In our strategy, the extent of dehydration plays a crucial role in the mechanical properties of hydrogels. The slightly-dehydrated LC hydrogel with $\phi_w = 70\%$ exhibits maximal fracture toughness. However, further dehydration leads to a decrease in toughness (Fig. 4b). For example, the toughness of LC hydrogels with $\phi_w = 55\%$ is 5248.7 J m$^{-2}$, much smaller than that of the LC hydrogels with $\phi_w = 70\%$. The reduction of toughness is attributed to a huge number of DIEs. The interchain entanglements are too dense. The $\alpha_{en}$ for an as-prepared LC hydrogel and a slightly-dehydrated LC hydrogel ($\phi_w = 70\%$) is 4.5 and 38.9, respectively (See Supplementary Fig. 11). With further dehydration to a water content of $\phi_w = 55\%$, the $\alpha_{en}$ increases rapidly to 110.7, indicating that the number density of entanglements becomes 110.7 times greater than that of the cross-linkers. Such dense entanglements take more time to remove, and some entanglements may not have sufficient time to unravel under a certain applied strain rate. Consequently, these entanglements that were not unraveled in time act as topologically physical cross-linkers. These additional physical cross-linkers reduce the stretchability of hydrogels (Fig. 4c). Furthermore, the entanglements that remain entangled during the deformation do not contribute significantly to energy dissipation, leading to a decrease in the energy dissipation ability (Fig. 4f). In addition, the size of FPZ decreases (See Supplementary Fig. 7). The fractocohesive length of LC hydrogels with water content $\phi_w = 55\%$ and $\phi_w = 40\%$ is $R_f = 7.8$ mm and $R_f = 0.95$ mm, respectively. Therefore, excessive DIEs, caused by heavy dehydration of LC hydrogels, are not conducive to achieving high toughness.

As mentioned in the introduction, so far, the majority of strategies developed to enhance both fracture toughness and fatigue threshold of hydrogels rely on constructing complex heterogeneous structures. Recently, Kim et al. achieved single network PAAm hydrogels with large modulus, high toughness, and large fatigue threshold[14]. While both their strategy and ours employ entanglements, the fundamental ideas of the two strategies differ significantly. Firstly, the recipe and preparation process are different. Kim et al. adopted a precursor solution with high monomer content (a monomer-to-water molar ratio of 0.5) and low cross-linker content (a cross-linker-to-monomer molar ratio of $1.0 \times 10^{-5}$). The precursor solution was polymerized to form a solid, and then the as-prepared hydrogel was adequately swollen. In contrast, we prepared a precursor solution with low monomer content (a monomer-to-water molar ratio of 0.04) and low cross-linker content (a cross-linker-to-monomer molar ratio of $5.0 \times 10^{-5}$). The precursor solution was polymerized, and the as-prepared hydrogel was slightly dehydrated. Secondly, the different recipes and preparation processes yield distinct microstructures. A precursor solution with crowded monomers results in a hydrogel with crowded polymers, thus, dense entanglements are formed during the polymerization. Most of these entanglements cannot be removed by swelling or external force unless chain scission occurs (Fig. 2c$_1$). In our strategy, most of the entanglements are introduced through the dehydration process, and these DIEs can be disentangled gradually when the external force increases (Fig. 2c$_2$ and c$_3$). Thirdly, the diverse microstructures further lead to various mechanical behaviors. The non-removable entanglements do not consume energy, making the corresponding hydrogel highly elastic with little hysteresis. The high toughness is endowed by the elastic dissipaters[14,46]. On the contrary, the DIEs dissipate energy considerably, thereby toughening our hydrogel. In summary, the hydrogel with dense non-removable PIEs exhibits larger strength, and these elastic entanglements offer advantages in applications such as sensing and actuating. By our strategy, the hydrogel with dense DIEs is highly crack-insensitive, with a much higher toughness (~22000 J m$^{-2}$) and a much larger fracture stretch (~20). The strong energy dissipation capability makes it suitable for applications requiring energy absorption.

Entanglements exist in most hydrogels. When dehydrated, the polymer chains get closer to each other, forming dehydration-induced entanglements. In this sense, the proposed strategy is generally applicable to most hydrogel systems. The current strategy was further applied to the polyacrylic acid (PAAc) hydrogel to verify its generality. Different from the PAAm hydrogel, the PAAc hydrogel features not only covalent cross-linking but also abundant physical interactions, such as hydrogen bonds, between the polymer chains. Aligned with the tested PAAm hydrogels, we prepared four types of PAAc hydrogels: the LC hydrogel *A* with $\phi_{w0} = 70\%$, the LC hydrogel *B* with $\phi_{w0} = 87\%$, the LC hydrogel *C* with an initial water content of $\phi_{w0} = 87\%$ and a current water content of $\phi_w = 70\%$, and a regular hydrogel with medium cross-linking and $\phi_{w0} = 87\%$. The measured fracture toughnesses, fatigue thresholds, and moduli of these hydrogels are listed in Supplementary Table 3, with the original experimental curves shown in Supplementary Fig. 12. The fracture toughness of a regular PAAc hydrogel is 116.8 J m$^{-2}$. By the current strategy, the toughness of hydrogel *C* increased 57.8 times to 6753.7 J m$^{-2}$, and the modulus and fatigue threshold improved by 3.0 times and 10.5 times in the meantime, respectively.

In the current strategy, the degree of entanglements can be conveniently modulated by the content of cross-linkers and the current water content of a hydrogel. For loosely cross-linked hydrogels, even slight dehydration leads to dense entanglements. The small reduction in water content preserves the advantage of high-water content of hydrogels. This paper provides an efficient strategy for the preparation of anti-fracture and fatigue-resistant hydrogels without the need for intricate microscopic design or processing. The designed hydrogel can undergo further modifications to enhance its stability, rendering it suitable for long-term applications[50–52]. One promising potential involves its utilization in wearable and washable conductors for active textiles. By incorporating hygroscopic salt into the hydrogel and then coated with a thin layer of butyl rubber, the resulting hydrogel exhibits excellent stability[53]. Consider the scenario where the hydrogel conductor is embedded into a textile, subsequently worn on the human body, or laundered in a washing machine. In this case, the hydrogel suffers from complex deformation and cyclic loading. Thus, to ensure the integrity of hydrogel conductors under both single and cyclic loading, high fracture toughness and fatigue threshold are necessary. The current strategy results in a PAAm hydrogel with a water content of 70%, a fracture stretch of more than 20, a fracture toughness of ~22,000 J m$^{-2}$, and a fatigue threshold of ~300 J m$^{-2}$, exhibiting potential application for the wearable and washable conductors for active textiles. The excellent overall mechanical performances endowed by our strategy will inspire researchers to explore the application of hydrogels in diverse research domains and under more demanding working conditions.

## Methods

### Materials preparation

Acrylamide (AAm, lot number A108465, AR 99.0%), $\alpha$-Ketoglutaric acid ($\alpha$-Keto, lot number K105570, AR 99.0%), and Acrylic acid (AAc, lot number A103526, AR 99.0%) were purchased from Aladdin. N,N′-Methylenebis(acrylamide) (MBAA, lot number M7279, AR 99.5%) was purchased from Sigma-Aldrich. All chemicals were used as received.

Polyacrylamide (PAAm) hydrogels were prepared from precursor solutions containing 86% mass fraction of water (i.e., 87% volume fraction of water, $\phi_{w0} = 87\%$). To obtain a medium cross-linked (MC) hydrogel, we combined 14 g of acrylamide powder (as the monomer), 1 ml of 0.1 mol L$^{-1}$ N,N′-Methylenebis(acrylamide) solution (as the

cross-linker), and 2 ml of 0.1 mol L$^{-1}$ $\alpha$-Ketoglutaric acid solution (as the initiator) in a conical tube. Deionized water was then added to achieve a total mass of 100 g for the solution. For low cross-linked (LC) hydrogels and high cross-linked (HC) hydrogels, the volume of the cross-linker solution was adjusted to 0.1 ml and 3 ml, respectively, while maintaining the weight of monomer powder and the volume of the initiator solution unchanged. The total mass of the precursor solution was kept at 100 g. Specifically, the cross-linker-to-monomer molar ratio for the LC/MC/HC PAAm hydrogel is $5.0 \times 10^{-5}$, $5.0 \times 10^{-4}$, and $1.5 \times 10^{-3}$, respectively. The initiator-to-monomer molar ratio for the LC/MC/HC PAAm hydrogel is kept at $1.0 \times 10^{-3}$. Polyacrylic acid (PAAc) hydrogels were prepared by adopting a similar process. The cross-linker-to-monomer molar ratio for the LC/MC PAAc hydrogel is $2.5 \times 10^{-4}$ and $1.0 \times 10^{-3}$, and the initiator-to-monomer molar ratio for the LC/MC PAAc hydrogel is kept at $1.0 \times 10^{-3}$. The precursor solution was stirred for 3 minutes without undergoing a degassing process due to the low viscosity of the solution. Next, the precursor solution was poured into molds and placed inside a UV light box for 3 hours to complete the polymerization.

Subsequently, the as-prepared hydrogels were either dehydrated by placing them in a low-humidity box or swollen in water. To prepare dehydrated hydrogel samples, we positioned the hydrogels on an acrylic plate, which was then placed inside a low-humidity box. The upper surface exposed to the air loses water quickly, while the lower surface in contact with the acrylic plate loses water very slowly, so the sample warps. To mitigate increasing warping, we inverted the upper and lower surfaces of the samples periodically. It should be mentioned that the polymer content (or water content) used in our text refers to the volume content, consistent with the convention of polymer physics. In experiments, we measured the weight of the hydrogels. To achieve a hydrogel with the desired polymer content $\phi_p$ (or water content $\phi_w$), the targeted weight of the hydrogel sample was calculated by $m = m_0 \psi_{p0}[(\phi_p^{-1} - 1)(\rho_w/\rho_p) + 1]$, where $m_0$ is the weight of the as-prepared hydrogel, $\psi_{p0}$ is the polymer content by weight of the as-prepared hydrogel ($\psi_{p0} = 14\%$ for hydrogels in the current research). The density of the polymer is labeled as $\rho_p$ and the density of water is labeled as $\rho_w$. The weights of the samples were monitored, and once the targeted weight was reached, the samples were removed and transferred into sealing bags for 72 hours. The sealing bags prevented water exchange between the hydrogels and the external environment, ensuring uniform water distribution within the hydrogel. Both optical micrographs (See Supplementary Fig. 13) and scanning electron microscope (SEM) images (See Supplementary Fig. 14) of the cross-sections validate the homogeneity of the hydrogel samples after 72 hours of sealing.

## Mechanical tests

**Uniaxial tension tests.** For the uniaxial tension tests, the as-prepared hydrogels with identical dimensions (a length of 80 mm, a width of 5 mm, and a thickness of 2 mm) were used to prepare dehydrated hydrogels. As a result, the hydrogel samples with various water contents for uniaxial testing own different sizes (See Supplementary Fig. 15). It should be noted that for uniaxial tension tests, the sample dimension does not matter, since the modulus, hyperelasticity, and viscoelasticity are size-independent.

For quasi-static uniaxial tension tests, the samples were stretched along the length direction at a low strain rate of $\dot{\varepsilon} = 0.001\,\text{s}^{-1}$ up to a stretch ratio of 5.0. To prevent water loss in the hydrogels during the long-term quasi-static tests, a test platform was constructed and utilized (See Supplementary Fig. 16). This platform consists of a mechanical test system (Instron 5965 with a force sensor of 10 N), along with a humidity control system comprising a humidity sensor, a humidity controller, and a humidifier. The relative humidity was kept

between 95% and 100%. For single-cycle tests, the samples were stretched along the length direction at a strain rate of $\dot{\varepsilon} = 0.1\,\text{s}^{-1}$ up to a maximal stretch ratio of 5.0 and then unloaded. The nominal stress is defined as the force divided by the initial cross-sectional area of the sample; the stretch ratio refers to the current length of the sample divided by its original length; the strain rate is calculated as the loading velocity divided by the original length of the sample. The elastic modulus is calculated from the linear region ($\lambda < 1.05$) of a s-s curve.

**Pure shear tests.** For pure shear tests, dehydrated hydrogels with identical dimensions (a width of 50 mm, a height of 10 mm, and a thickness of about 2 mm) were employed in the mechanical tests. We designed molds with various sizes for the preparation of initial hydrogels, ensuring that dehydrated samples with various water contents maintain uniform size (See Supplementary Fig. 17). Some samples were precut with a 20 mm-long crack before testing. Both the uncut and precut samples were stretched along the height direction at a strain rate of $\dot{\varepsilon} = 0.1\,\text{s}^{-1}$ until they fractured. The fracture stretches of uncut/precut samples were labeled as $\lambda_f$ and $\lambda_c$, respectively. The area underneath the nominal stress-stretch ratio (s-s) curve of uncut samples stands for the strain energy density $W(\lambda)$ of a deformed hydrogel with a stretch ratio of $\lambda$. The strain energy density $W(\lambda_f)$ refers to the work of fracture. The fracture toughness is calculated as $W(\lambda_c)H$, with $H$ being the reference height of the sample. The fractocohesive length is defined as $W(\lambda_c)H/W(\lambda_f)$. For some samples with $\lambda_f < \lambda_c$ (e.g., the LC hydrogels with $\phi_w = 87\%$), we considered the value of $W(\lambda_f)H$ as the fracture toughness. Larger samples with a height of 40 mm were adopted in pure shear tests for LC hydrogels with water contents of 87% and 70%.

The fatigue tests adopt the same samples as the toughness tests. A platform was constructed to minimize the change in water content of hydrogel samples during the fatigue test (See Supplementary Fig. 18). The platform consists of a mechanical test system, a humidity control system, and a sealing box. It is worth mentioning that the sealing box comprises an acrylic box (the lower part without a cover plate) and a removable acrylic cover plate (the upper part). The sealing box is designed with two independent parts for two reasons. First, the cover plate can be removed to facilitate the installation of hydrogel samples. Second, the relative humidity within the sealing box is very high. As the experimental time increases, a layer of water droplets will form on the surface of the cover plate which causes trouble in photographing the crack tip of hydrogel samples. To solve this problem, in the experiment, we took off the cover plate and wiped off the water droplets quickly. The hydrogel sample was placed within the sealing box and the relative humidity was kept between 95% to 100%. A precut sample was cyclically loaded with a maximal stretch ratio of $\lambda_m$, and the crack growth process was recorded using a digital camera. For every 1000 cycles, we recorded a 10-second movie. The photos corresponding to the maximal deformation were extracted to analyze the location of the crack. The crack extension was plotted against the number of cycles. A steady state was observed after many cycles, when the crack extension was almost linear with respect to the number of cycles. The steady region was adopted to calculate the crack extension per cycle (d$c$/d$N$). An uncut sample was also cyclically stretched to measure the corresponding energy release rates. For each $\lambda_m$, the loading curve of the 100th cycle was taken to calculate the energy release rate. The area underneath the 100$^\text{th}$ loading s-s curve represented the strain energy density $W(\lambda_m)$. The energy release rate $G$ was calculated as $W(\lambda_m)H$. The curve of the crack extension per cycle (d$c$/d$N$) with respect to the energy release rate $G$ was plotted. In cases of slow crack growth, the crack extension per cycle (d$c$/d$N$) varied basically linearly with energy release rate $G$. These data points were fitted with a line that intercepts the $G$ axis. This intercept was taken as the fatigue threshold.

## Data availability

The data that supports the findings of this work can be found in the Main Text and the Supplementary Information. Data is available from the authors on request.

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

## Acknowledgements

The authors thank Dr. Tenghao Yin for discussions and comments on the manuscript. The authors acknowledge the following funding support: National Natural Science Foundation of China (Nos. 12321002 (S.Q.), 12202397 (D.Z.), 12132014 (S.Q.), 12022204 (R.X.)); 111 Project of China (No. B21034 (S.Q.)); Zhejiang Provincial Natural Science Foundation of China (No. LD22A020001 (R.X.)); China Postdoctoral Science Foundation (No. 2022M722823 (D.Z.)).

## Author contributions

Conceptualization: D.Z. and S.Q. Methodology: D.Z., Z.W., J.L., and R.X. Validation: D.Z., Z.W., J.X., and R.X. Supervision: S.Q. and W.Y. Writing-original draft: D.Z. Writing-review & editing: D.Z., Z.W., S.Q., and W.Y.

## Competing interests

The authors declare no competing interests.
