## [Peer Review File · Nature Communications]

A strategy for tough and fatigue-resistant hydrogels via loose cross-linking and dense dehydration-induced entanglementsREVIEWER COMMENTS

Reviewer #1 (Remarks to the Author):

This work presented that the dehydration-induced entanglements of the polymer chains in the hydrogel with low crosslinking density led to overcoming the stiffness-toughness trade-off of the single-chain hydrogel. The experimental design of the hydrogel with different crosslinking density and water contents is well proposed to compare its mechanical properties. The introduction was well written, and the overall data analysis and discussion are clear to support the central concept of this work. However, here are a few points to be further considered, including the consistency of water contents over the mechanical measurements and the practical use of the proposed slightly dehydrated hydrogel in the intermediate state.

- In Figure 3, the hydrogel's lowest water content is 40%. Is there any reason not to study the hydrogel with less than 40% water content?

- In general, different water contents in the hydrogel are closely related to the size or volume of the hydrogel. It is unclear whether the dehydrated hydrogels with the same dimension were used in the mechanical test or whether the initial hydrogels with the same dimension were used to prepare dehydrated gels. It would be very helpful for the reader to have more detailed experimental to prepare the dehydrated hydrogels and the photographs of the dehydrated hydrogels tested.

- In Fig 5e, it seems that the hydrogel was exposed to open air during the cyclic test, which can cause additional dehydration and affect the gel's mechanical properties. What are the water contents of the hydrogel before and after the 5,000 and 20,000 cyclic tests shown in Fig 5f? This would support the idea that the mechanical property of the measured hydrogel is attributed to the DIEs rather than possible dehydration during the test.

- The authors described dehydrating the as-prepared hydrogels by placing them in a low-humidity box. Typically, dehydration occurs from the hydrogel's outer surface, and when the designated water contents are achieved by measuring the gel's weight, the hydrogel is retrieved. In this case, how is the uniformity of the DIEs of hydrogel throughout the cross-section? It is suggested that the cross-sectioned SEM images of the hydrogel be presented.

- In the previous paper (ref. 14), the as-prepared hydrogel with very low water content was fully swollen in water to reach equilibrium, resulting in a highly entangled hydrogel. In contrast, in this work, the final form of the hydrogels is in a dehydrated/intermediate state far from equilibrium. From a practical point of view, this intermediate state of the hydrogel would be a disadvantage because the mechanical properties will be changed depending on the level of dehydration. Is there any potential application using this gel? Is there any possible way to make this intermediate state to a more stable hydrogel?

Reviewer #2 (Remarks to the Author):

In this manuscript, the authors proposed a facile strategy to enhance the fracture toughness,

fatigue threshold, and stiffness of hydrogels with single network structure. By the design of loose cross-linking with slight dehydration, the mechanical properties of polyacrylamide hydrogel are greatly improved. The experiments are designed rationally and the results are presented well, which make the paper a convincing piece of work. Several issues listed below should be considered:

1. It's notable that in this paper the fracture toughness of the hydrogel seems to increase with the sample size, even reaching about 22000 J/m². Why the behaviors of larger samples differ from that of smaller ones and doesn't align with the pure shear test standard?
2. The dehydration process appears to result in varying degrees of water loss across different areas in a hydrogel. How to achieve uniform dehydration and prevent the sample from warping?
3. Dehydration can significantly impact the mechanical properties of hydrogels, especially in loosely crosslinked hydrogels. How to accurately control the water content of hydrogel and avoid excessive water loss or swelling during the experimental process, particularly during fatigue testing? would the hydrogels with loose crosslinking and controlled dehydration be susceptible to ambient humidity when under applications?
4. In Figure 4b, the fracture toughness of the LC hydrogel exhibits a unique trend which first increase and then decrease with the reduction of water content. This is quite different from the trends observed for hydrogels with other cross-linking degrees. Could you provide further clarification on this?
5. Figure 4c presents an interesting result that the cracked sample with an initial water content of 87% exhibits a larger fracture stretch ratio. Is there any explanation for this phenomenon?
6. During fatigue testing, it's assumed that the stress-strain curve reaches a steady state after 100 cycles. However, the maximum stress in Figure 5c seems to continue decreasing. If the load is sustained, would the stress-strain curve continue to decline? It might be beneficial to present cyclic curves over more cycles to provide a more comprehensive understanding of the material's fatigue behavior over an extended period.
7. If a hydrogel is prepared directly with water content of 70%, bypassing the dehydration process, would the result be same?

Point-to-point response to reviewers' comments

We would like to thank the reviewers for their thoughtful comments and suggestions, which can significantly improve the quality of our work. Changes in the manuscript are highlighted in red. The following are our point-to-point responses to their comments:

Response to Reviewer 1's comments:

Reviewer 1 (Remarks to the Author):

This work presented that the dehydration-induced entanglements of the polymer chains in the hydrogel with low crosslinking density led to overcoming the stiffness-toughness trade-off of the single-chain hydrogel. The experimental design of the hydrogel with different crosslinking density and water contents is well proposed to compare its mechanical properties. The introduction was well written, and the overall data analysis and discussion are clear to support the central concept of this work. However, here are a few points to be further considered, including the consistency of water contents over the mechanical measurements and the practical use of the proposed slightly dehydrated hydrogel in the intermediate state.

1. In Figure 3, the hydrogel's lowest water content is 40%. Is there any reason not to study the hydrogel with less than 40% water content?

Answer:

Thank you for this question. In our experimental study, the water content range was set from 40% to 93%. The hydrogels with lower water contents were not further investigated because of the following reasons. First, for hydrogel materials, a decrease in water content correlates with an increase in the glass transition temperature^[1]. The mechanical behaviors of hydrogels with a water content of 40% indicate that they still belong to the rubbery state at room temperature (e.g., no abrupt increment of modulus was observed when the water content decreased gradually to 40%). While the further decreasing of water content may transition the hydrogel into a glassy state at room temperature. In the current work, we focus on exploring the effect of dehydration-induced entanglements on the mechanical properties of hydrogels, rather than investigating the effects of glass transition. Second, hydrogels with further dehydration may lose the advantage of high water content which is an important merit of hydrogel. Consequently, hydrogels with water contents below 40% were not further studied here.

[1] Xiao, R., Li, H., 2021, Glass transition in gels, Phys. Rev. Mater., 5, 065604

2. In general, different water contents in the hydrogel are closely related to the size or

volume of the hydrogel. It is unclear whether the dehydrated hydrogels with the same dimension were used in the mechanical test or whether the initial hydrogels with the same dimension were used to prepare dehydrated gels. It would be very helpful for the reader to have more detailed experimental to prepare the dehydrated hydrogels and the photographs of the dehydrated hydrogels tested.

Answer:

We thank the reviewer for these helpful suggestions, and we agree that more experimental details about the hydrogel samples should be added.

As highlighted by the reviewer, the size of the hydrogel sample changes with the water content. In our experimental setup, for the uniaxial tension tests, the as-prepared hydrogels with the same dimension were used to prepare dehydrated hydrogels. As a result, the hydrogel samples with various water contents for uniaxial testing own different sizes (See Fig. R1). It should be noted that for uniaxial tension tests, the sample dimension does not matter, since the modulus, hyperelasticity, and viscoelasticity are size-independent.

For the pure shear tests, the dehydrated hydrogels with the same dimension were used in the mechanical test. We designed molds with various sizes for the preparation of initial hydrogels so that the dehydrated hydrogel samples with various water contents own the same size (See Fig. R2). The fracture toughness of a hydrogel is size-dependent. Below the fractocohesive length, larger sample dimensions correspond to higher measured toughness due to increased energy dissipation around the crack, contributing to fracture toughness. Therefore, the sample dimension of the pure shear tests for hydrogels with various water contents was kept consistent.

We have added photographs of the initial/dehydrated hydrogel samples in the Supplementary Information (Supplementary Figs. 15 and 17), and corresponding illustrative text was included in the Methods section of the manuscript (Methods - Mechanical tests).

Fig. R1. Photos of PAAm hydrogel samples for uniaxial tension tests. (A) The as-prepared hydrogels with an initial water content of 87% have the same dimensions. (B) The dehydrated hydrogels with various current water contents. The lower the water content, the smaller the sample size.

Fig. R2. Photos of PAAm hydrogel samples for pure shear tests. (A) The as-prepared hydrogels with an initial water content of 87% were designed with various initial sizes. The lower the targeted water content, the larger the size of the as-prepared hydrogel sample. (B) The dehydrated hydrogels with various current water contents have the same dimensions.

3. In Fig 5e, it seems that the hydrogel was exposed to open air during the cyclic test, which can cause additional dehydration and affect the gel's mechanical properties. What are the water contents of the hydrogel before and after the 5,000 and 20,000 cyclic tests shown in Fig 5f? This would support the idea that the mechanical property of the measured hydrogel is attributed to the DIEs rather than possible dehydration during the test.

Answer:

We apologize that the detailed experimental setting of the fatigue tests was not provided in the manuscript. In fact, as shown in Fig. R3, we constructed a platform to minimize the change in water content of hydrogel samples during the fatigue test.

The platform consists of the mechanical test system, the humidity control system, and the sealing box. It is worth mentioning that the sealing box comprises an acrylic box (the lower part without a cover plate) and a removable acrylic cover plate (the upper part). The sealing box is designed with two independent parts for the following reasons. First, the cover plate can be removed to facilitate the installation of hydrogel samples. Second, the relative humidity within the sealing box is very high. As the experimental time increases, a layer of water droplets will form on the surface of the cover plate which causes trouble in photographing the crack tip of hydrogel samples. To solve this problem, in the experiment, we removed the cover plate and wiped off the water droplets quickly. Fig. 5e in the manuscript exhibited the top view, giving the impression that the sample was exposed to open air. We have added both the front view and the top view of the fatigue test platform in the Supplementary Information (Supplementary Fig. 18).

For fatigue tests, the hydrogel sample was placed within the sealing box and the relative humidity was kept between 95% to 100%. We believe that at such high relative humidity, the volatilization of water within the hydrogel is negligible. Following the reviewer's suggestion, we measured the mass change of hydrogel samples after 5000 and 20000 cyclic tests. The LC hydrogel with a current water content of 70% was adopted. The maximal stretch ratio was set as $\lambda=2.5$. The initial mass of the hydrogel sample is 3.139 g, and the masses after 5000 and 20000 cycles are 3.144 g and 3.215 g, respectively. The water contents after 5000 and 20000 cyclic tests were measured as 70.05% and 70.71%, respectively. The experimental results indicate that the change of water content during the fatigue test is negligible, and the mechanical property of the measured hydrogel is attributed to the designed DIEs rather than the further dehydration during the test. The mass of the hydrogel sample increases slightly because a tiny amount of atomized water moved onto the surface of the hydrogel and was absorbed by the hydrogel sample directly.

Fig. R3. Experimental setting of the fatigue test of hydrogel samples.

4. The authors described dehydrating the as-prepared hydrogels by placing them in a low-humidity box. Typically, dehydration occurs from the hydrogel's outer surface, and when the designated water contents are achieved by measuring the gel's weight, the hydrogel is retrieved. In this case, how is the uniformity of the DIEs of hydrogel throughout the cross-section? It is suggested that the cross-sectioned SEM images of the hydrogel be presented.

Answer:

We think the uniformity of the dehydration-induced entanglements (DIEs) is an essential problem. The uniformity of the DIEs relates to the homogeneity of the water content directly. If the water content distributes uniformly through the hydrogel sample, then the DIEs distribute uniformly too. As shown in Fig. R4A, the cross-section of an as-prepared hydrogel with a water content of 87% is a rectangle (The four vertices of the cross-section have a certain curvature, due to the limitations of the casting method).

As the reviewer mentioned, dehydration occurs from the hydrogel's outer surface. Once the designated water content (i.e., the designated mass of the hydrogel sample) was reached, the dehydration process was stopped. At this moment, we cut the sample to show the cross-section (Fig. R4B). Obviously, the cross-section looks like an ellipse, the original two straight long sides become two curves. The reason is that the distribution of water content through the sample is quite nonuniform, though the average water content reaches the target value. To make both the water content and the DIEs uniform through the hydrogel, before the testing of hydrogel samples, we transferred the dehydrated samples into sealing bags for 72 hours. The water molecules diffuse from the high water content region to the low water content region, making the distribution of water content uniform. As exhibited in Fig. R4C, after 72 hours of sealing, the cross-section of the dehydrated hydrogel turns back to a rectangle, and the two long sides become straight lines, indicating the uniformity of water content

distribution.

Fig. R4. Optical micrographs of cross-sections of hydrogel samples. (A) As-prepared hydrogel with a water content of 87%. (B) Hydrogel that has just been dehydrated to a water content of 40%. (C) The dehydrated hydrogel that has been placed in a sealing bag for 72 hours.

Following the reviewer’s suggestion, the SEM images of a dehydrated hydrogel sample with a water content of 40% are also provided. The dehydrated hydrogel was placed in a sealing bag for 72 hours to uniform the distribution of water content. Subsequently, the sample was freeze-dried for 48 hours before the characterization of SEM. Fig. R5 exhibits the images at three various locations on the cross-section, and similar microscopic structures can be observed, with the size of the “mesh” being around 300-500 nm. The uniformity of the polymer network structure indicates that the distribution of water content through the dehydrated hydrogel should be uniform.

Fig. R5. SEM images of a dehydrated hydrogel sample with a water content of 40%. Microscopic structures at various locations at the cross-section are similar.

We have added optical micrographs and SEM images in the Supplementary

Information (Supplementary Figs. 13 and 14), and corresponding illustrative text was included in the Methods section of the manuscript (Methods - Materials preparation).

5. In the previous paper (ref. 14), the as-prepared hydrogel with very low water content was fully swollen in water to reach equilibrium, resulting in a highly entangled hydrogel. In contrast, in this work, the final form of the hydrogels is in a dehydrated/intermediate state far from equilibrium. From a practical point of view, this intermediate state of the hydrogel would be a disadvantage because the mechanical properties will be changed depending on the level of dehydration. Is there any potential application using this gel? Is there any possible way to make this intermediate state to a more stable hydrogel?

Answer:

We thank the reviewer for these two questions. The first question concerns the potential application of this hydrogel, and the second question is about the possible methods to make this dehydrated hydrogel a more stable one. We'd like to answer the second question first because once this intermediate hydrogel can be designed into a stable material, this gel will have a wide range of applications.

5.1. Is there any possible way to make this intermediate state to a more stable hydrogel?

A common hydrogel is unstable in the ambient condition. It tends to lose water when applied directly for a long time and its mechanical properties and other corresponding properties change in the meantime. Typically, there are three types of methods to make the hydrogel a more stable material.

The first method to make a more stable hydrogel is by coating another layer of soft material onto the surface of a hydrogel ^[1]. The coated material greatly suppresses the exchange of water molecules between the hydrogel and the external environment, thus making the hydrogel material stable. The coating method is simple and versatile and can be applied to a wide range of hydrogels ^[2]. The coated soft material can be diverse commonly used elastomers (e.g., polydimethylsiloxane Sylgard 184, Latex, VHB, and Ecoflex) ^[3], hydrophobic polymer ^[4], and hydrophilic substance ^[1]. The interface between the hydrogel and the coated soft material can be designed to be robust ^[3]. Moreover, the coated layer is thin and the coated soft material can also be highly stretchable, therefore, the coating method does not compromise the mechanical properties of hydrogels. For example, a polyacrylamide hydrogel with a 200- μm thick double-hydrophobic-coating exhibits strong resistance to both drying in air and swelling in aqueous environments ^[5]. The mass of coated hydrogel sample remains basically unchanged either placed in air or placed underwater for 7 days.

The second type of method to improve the water retention ability is introducing hygroscopic salt into the hydrogel. The dissociation of hygroscopic salts in water results in the formation of hydrated ions, decreasing the vapour pressure ^[6]. The presence of hygroscopic ions effectively strengthens the interactions between the water molecules

within the hydrogel, preventing water evaporation ^[7]. The commonly adopted salts include lithium chloride, potassium acetate, magnesium chloride, and calcium chloride, among which lithium chloride is the most efficient one ^[8,9]. For example, a polyacrylamide hydrogel containing a high content of lithium chloride retains water at an extremely low relative humidity of 10% ^[10]. Moreover, the addition of hygroscopic ions transfers the hydrogel into a conductive material, making it a powerful candidate for applications such as wearable bioelectronic devices, ionic skins, and flexible electronics.

In the third method, the hydrogel is transferred to the organohydrogel by partially replacing water with hygroscopic solvents (e.g., glycerol, glycol, and sorbitol). The strong cooperative hydrogen bonding between the organic solvents and water molecules firmly anchored the water in the polymer network, and therefore endowed the organohydrogel with the properties of long-term stability ^[11]. The organohydrogel can be either directly polymerized in a glycerol–water binary solvent ^[12], or be completed by a facile solvent-exchange method ^[13]. A hydrogel becomes non-drying when some water is displaced with hygroscopic solvents. For example, an organohydrogel is fabricated by immersing a tough hydrogel in the glycerol solvent for three hours. The mass of the resulting glycerol-hydrogel was kept almost the same when it was placed at 20 °C and 50% relative humidity for 8 days ^[14].

The above-mentioned methods can be combined to achieve better stability of hydrogels. For instance, the combination of hygroscopic salt and hydrophobic coating eliminates dehydration ^[15].

We analyze the feasibility of combining the above methods with the strategy developed in this paper. In the current strategy, the hydrogel is loosely cross-linked and slightly dehydrated. The dehydrated hydrogel can be coated with elastomers and polymers just like the as-prepared hydrogels, therefore, the coating method is feasible. Hygroscopic salts can be added into the precursor of hydrogels. Since the salt concentration increases with dehydration, less salt is needed in the precursor solution. The current strategy is possible to be extended to the organohydrogels. An as-prepared hydrogel is dehydrated first, then some of the remaining water is replaced with hygroscopic solvent. The resulting organohydrogel has a long chain structure and dehydration-induced entanglements, and it should have excellent anti-fracture and anti-fatigue ability, similar to the highly entangled hydrogel. In the current work, we focus on the hydrogel materials, and extending the current strategy to organohydrogels/organogels will be the focus of future work. The three methods to make a hydrogel more stable are summarized in Table. R1.

Table. R1 Analyzation of the combination of the proposed strategy and some methods to make a hydrogel more stable

Methods	Feasibility	Anti-drying	Anti-swelling in solution
Coating	Feasible	Yes	Yes
Adding hygroscopic salt	Feasible	Yes	No
Organohydrogel	Possible	Yes	No

- [1] Gao, W., Chang, J., Li, X., Li, S., Zhou, Y., Hou, X., Long, L., Zhao, J., Yuan, X., 2023, A Quenched Double-Hydrophilic Coating for the Enhancement of Water Retention of Hydrogels, *Adv. Funct. Mater.*, 33, 2303306
- [2] Li, X., Bian, F., Shi, J., Zhang, E., Kong, C., Ren, J., Wu, K., 2023, A novel environment-tolerant hydrogel via a combination effect of a polyurethane coating and hygroscopic salt for underwater monitoring, *J. Mater. Chem. C.*, 11, 5388-5401
- [3] Yuk, H., Zhang, T., Parada, G., Liu, X., Zhao, X., 2016, Skin-inspired hydrogel-elastomer hybrids with robust interfaces and functional microstructures, *Nat. Commun.*, 7, 12028
- [4] Zhu, T., Jiang, C., Wang, M., Zhu, C., Zhao, N., Xu, J., 2021, Skin - inspired double - hydrophobic - coating encapsulated hydrogels with enhanced water retention capacity, *Adv. Funct. Mater.*, 31, 2102433
- [5] Mredha, M., Le, H., Cui, J., Jeon, I., 2020, Double - Hydrophobic - Coating through Quenching for Hydrogels with Strong Resistance to Both Drying and Swelling, *Adv. Sci.*, 7, 1903145
- [6] Young, J., 1967, Humidity control in the laboratory using salt solutions—a review, *J. App. Chem.*, 17, 241-245
- [7] Ge, W., Cao, S., Yang, Y., Rojas, O., Wang, X., 2021, Nanocellulose/LiCl systems enable conductive and stretchable electrolyte hydrogels with tolerance to dehydration and extreme cold conditions, *Chem. Eng. J.*, 408, 127306
- [8] Sui, X., Guo, H., Cai, C., Li, Q., Wen, C., Zhang, X., Wang, X., Yang, J., Zhang, L., 2021, Ionic conductive hydrogels with long-lasting antifreezing, water retention and self-regeneration abilities, *Chem. Eng. J.*, 419, 129478
- [9] Tang, L., Wu, S., Li, Y., Jiang, K., Xu, Y., Dai, B., Wang, W., Tang, J., Gong, L., 2023, A super-tough ionic conductive hydrogel with anti-freezing, water retention, and self-regenerated properties for self-powered flexible sensor, *Appl. Mater. Today.*, 32, 101820
- [10] Bai, Y., Chen, B., Xiang, F., Zhou, J., Wang, H., Suo, Z., 2014, Transparent hydrogel with enhanced water retention capacity by introducing highly hydratable salt, *Appl Phys Lett*, 105, 151903
- [11] Han, L., Liu, K., Wang, M., Wang, K., Fang, L., Chen, H., Zhou, J., Lu, X., 2018, Mussel - inspired adhesive and conductive hydrogel with long - lasting moisture and extreme temperature tolerance, *Adv. Funct. Mater.*, 28, 1704195
- [12] Carvalho, F., Lopes, P., Carneiro, M., Serra, A., Coelho, J., Almeida, A., Tavakoli, M., 2020, Nondrying, sticky hydrogels for the next generation of high-resolution conformable bioelectronics, *ACS Appl. Electron. Mater.*, 2, 3390-3401
- [13] Wu, J., Wu, Z., Xu, H., Wu, Q., Liu, C., Yang, B., Gui, X., Xie, X., Tao, K., Shen, Y., 2019, An intrinsically stretchable humidity sensor based on anti-drying, self-healing and transparent organohydrogels, *Mater. Horiz.*, 6, 595-603
- [14] Chen, F., Zhou, D., Wang, J., Li, T., Zhou, X., Gan, T., Wang, S., Zhou, X., 2018, Rational fabrication of anti - freezing, non - drying tough organohydrogels by one-pot solvent displacement, *Angew. Chem. Int. Ed. Engl.*, 130, 6678-6681
- [15] Floch, P., Yao, X., Liu, Q., Wang Z., Nian G., Sun, Y., Jia, L., Suo, Z., 2017, Wearable and washable conductors for active textiles, *ACS Appl. Mater. Interfaces*, 9, 25542-25552

5.2. Is there any potential application using this gel?

Employing the current strategy yields a single network hydrogel endowed with exceptional stretchability, ultra-high fracture toughness, a high fatigue threshold, remarkable crack insensitivity, and relatively high water content. These outstanding mechanical properties and the strategy's simplicity position the designed hydrogel as a promising candidate for numerous applications. Some examples are given below.

The PAAm hydrogel exhibited in this work can be further transformed into conductive hydrogels through various methods, including the addition of conductive salts (e.g., NaCl) ^[1], graphene ^[2] or substituting water with conductive solvents (e.g., deep eutectic solvent) ^[3]. Employed as artificial skin, the hydrogel is attached to the surface of human skin to function as sensors, such as deformation and pressure sensors ^[4]. When monitoring the finger bending, wrist bending, and knee movement, the hydrogel experiences large deformation, necessitating excellent stretchability. Moreover, for prolonged monitoring, the artificial skin endures cyclic loading, demanding high anti-fatigue properties of the hydrogel. The hydrogels we developed effectively meet the above mechanical property requirements.

This hydrogel holds potential for applications in wearable and washable conductors for active textiles. Floch et al developed a wearable and washable hydrogel conductor ^[5]. Hygroscopic salt was incorporated into the hydrogel, and the hydrogel surface was coated with a thin layer of butyl rubber. The designed hydrogel fiber exhibits excellent stability. For instance, the hydrogel fiber exhibited little mass variation even after exposure to air for 50 days. After soaking in water for 5 days, the conductance remains basically unchanged, indicating that there is no obvious leakage of salts within the hydrogel. Moreover, they further washed the hydrogel fiber using a washing machine. They went up to 5 cycles of washing and each washing took 35 minutes. Imagine that when the hydrogel conductor is embedded into a textile and then washed by a washing machine, the hydrogel suffers from various complex deformations, such as stretching, compression, bending, torsion, and the combination of these deformation modes. What's worse, the hydrogel conductor embedded in a textile may undergo hundreds of washing during its service, and each washing may cause thousands of cyclic loading. In addition, the hydrogel conductor also experiences fatigue loading when the textile is worn on the human body. Thus, to ensure the integrity of hydrogel conductors under both single and cyclic loading, high fracture toughness and fatigue threshold are necessary. The current strategy results in a PAAm hydrogel with a water content of 70%, a fracture stretch of more than 20, a fracture toughness of $\sim 10,000 \text{ J/m}^2$, and a fatigue threshold of $\sim 300 \text{ J/m}^2$, indicating potential application for the wearable and washable conductors for active textiles.

In addition to the possible applications in artificial skins and wearable, washable conductors for active textiles, more potential applications include soft robotics ^[6-7], flexible electronics ^[8-9], and beyond, all demanding outstanding overall mechanical properties.

The analysis of potential applications has been added to the last paragraph of the

Discussion section of the manuscript.

[1] Sun, J., Keplinger, C., Whitesides, G., Suo, Z., 2015, Ionic skin, *Adv. Mater.*, 26, 7608-7614

[2] Xue, B., Sheng, H., Li, Y., Li, L., Di, W., Xu, Z., Ma, L., Wang, X., Jiang, H., Qin, M., Yan, Z., Jiang, Q., Liu, J., Wang, W., Cao, Y., 2021, Stretchable and self-healable hydrogel artificial skin, *Natl. Sci. Rev.*, 9, nwab147

[3] Zhang, Y., Wang, Y., Guan, Y., Zhang, Y., 2022, Peptide-enhanced tough, resilient and adhesive eutectogels for highly reliable strain/pressure sensing under extreme conditions, *Nat. Commun.*, 13, 6671

[4] Hu, L., Chee, P., Sugiarto, S., Yu, Y., Shi, C., Yan, R., Yao, Z., Shi, X., Zhi, J., Kai, D., Yu, H., Huang, W., 2023, Hydrogel-based flexible electronics, *Adv. Mater.*, 35, 2205326

[5] Floch, P., Yao, X., Liu, Q., Wang, Z., Nian, G., Sun, Y., Jia, L., Suo, Z., 2017, Wearable and washable conductors for active textiles, *Acs Appl. Mater. Inter.*, 9, 25542-25552

[6] Lee, Y., Song, W., Sun, J., 2020, Hydrogel soft robotics, *Mater. Today. Phys.*, 15, 100258

[7] Chen, Y., Zhang, Y., Li, H., Shen, J., Zhang, F., He, J., Lin, J., Wang, B., Niu, S., Han, Z., Guo, Z., 2023, Bioinspired hydrogel actuator for soft robotics: Opportunity and challenges, *Nano. Today.*, 49, 101764

[8] Keplinger, C., Sun, J., Foo, C., Rothmund, P., Whitesides, G., Suo, Z., 2013, Stretchable, Transparent, Ionic Conductors, *Science*, 341, 984-987

[9] Yang, C., Suo, Z., 2018, Hydrogel ionotronics, *Nat. Rev. Mater.*, 3, 125-142

Response to Reviewer 2's comments:

Reviewer 2 (Remarks to the Author):

In this manuscript, the authors proposed a facile strategy to enhance the fracture toughness, fatigue threshold, and stiffness of hydrogels with single network structure. By the design of loose cross-linking with slight dehydration, the mechanical properties of polyacrylamide hydrogel are greatly improved. The experiments are designed rationally and the results are presented well, which make the paper a convincing piece of work. Several issues listed below should be considered.

1. It's notable that in this paper the fracture toughness of the hydrogel seems to increase with the sample size, even reaching about 22000 J/m². Why the behaviors of larger samples differ from that of smaller ones and doesn't align with the pure shear test standard?

Answer:

Yes, for the low cross-linked (LC) hydrogels with relatively high water contents (87% and 70%), the larger sample size results in larger fracture toughness.

The observed phenomenon can be attributed to the extremely large fractocohesive length of these hydrogels. Taking the LC hydrogel with a water content of 70% as an example, the fractocohesive length is measured as 28.5 mm. The fractocohesive length characterizes the size of the fracture process zone (FPZ), and the energy dissipated in the FPZ contributes to the fracture toughness^[1]. The size of the FPZ for larger samples is 28.5 mm, whereas for smaller samples, it is confined to 10 mm, limited by the sample's height (See Fig. R6). With more materials to dissipate energy, the measured toughness of the larger samples is much higher.

Similar phenomena have been observed in the measurement of toughness by the 90-degree peeling test^[1], 180-degree peeling test^[2], and tearing test^[3]. However, the size-dependent fracture toughness values were rarely reported when the pure shear tests were adopted, because the fractocohesive length of most hydrogel materials (no more than a few millimeters)^[4] is smaller than the height of the sample (10 mm).

Fig. R6. Schematic diagram for the fracture process zone (FPZ) of a large sample and a small sample. The FPZ in the larger sample is much larger than that in the smaller sample, thus resulting in a higher fracture toughness of the larger sample.

- [1] Liu, J., Yang, C., Yin, T., Wang, Z., Qu, S., Suo, Z., 2019, Polyacrylamide hydrogels. II. elastic dissipater, *J. Mech. Phys. Solids.*, 133, 103737
- [2] Yin, T., Zhang, G., Qu, S., Suo, Z., 2021, Peel of elastomers of various thicknesses and widths, *Extreme Mech. Lett.*, 46, 101325
- [3] Jia, Y., Zhou, Z., Jiang, H., Liu, Z., 2022, Characterization of fracture toughness and damage zone of double network hydrogels, *J. Mech. Phys. Solids.*, 169, 105090
- [4] Chen, C., Wang, Z., Suo, Z., 2017, Flaw sensitivity of highly stretchable materials, *Extreme Mech. Lett.*, 10, 50-57

2. The dehydration process appears to result in varying degrees of water loss across different areas in a hydrogel. How to achieve uniform dehydration and prevent the sample from warping?

Answer:

Thank you for this question. To prepare dehydrated hydrogel samples, we position the hydrogels on an acrylic plate, which is then placed inside a low-humidity box. The upper surface exposed to the air loses water quickly, while the lower surface in contact with the acrylic plate loses water very slowly, so the sample warps. To mitigate increasing warping, we inverted the upper and lower surfaces of the samples periodically. In this way, the dehydrated samples warp slightly. After further placing the dehydrated hydrogels in sealing bags for 72 hours, the water content within the hydrogels becomes uniformly distributed, and the warping disappears.

Some relative experimental details were added to the Methods section of our manuscript (Methods - Materials preparation).

3. Dehydration can significantly impact the mechanical properties of hydrogels, especially in loosely crosslinked hydrogels. How to accurately control the water content of hydrogel and avoid excessive water loss or swelling during the experimental process, particularly during fatigue testing? Would the hydrogels with loose crosslinking and controlled dehydration be susceptible to ambient humidity when under applications?

Answer:

We believe the reviewer's questions are important. It is necessary to keep the water content during both experimental testing and applications.

For the first question, we constructed platforms for the quasi-static loading tests (see Fig. R7) and the fatigue tests (see Fig. R8). Hydrogel samples were put into an acrylic box where the relative humidity was controlled between 95% to 100%. At such a high relative humidity, the water loss of hydrogels is really slight. Following another reviewer's suggestion, we measured the change of water content for samples of fatigue tests. It was observed that the change of water content (from 70% to 70.71%) after 20000 cycles is negligible. The mass of the hydrogel sample increases slightly because

a tiny amount of atomized water moved onto the surface of the hydrogel and was absorbed by the hydrogel sample directly.

Photos of the experimental setting of the quasi-static test and fatigue test have been added in the Supplementary Information (Supplementary Figs. 16 and 18). The corresponding illustrative text was added in the Methods section of the manuscript (Methods - Mechanical tests).

Fig. R7. Experimental setting of the quasi-static test of hydrogel samples.

Fig. R8. Experimental setting of the fatigue test of hydrogel samples.

The second question is about the stability of hydrogels, which is also a concern of another reviewer. We think the designed hydrogel should be susceptible to ambient humidity if applied without any special treatment for a long term. The hydrogel will lose water under ambient humidity. Therefore, methods to keep the water content are necessary. Typically, there are three types of methods to make the hydrogel a more stable material.

The first method to make a more stable hydrogel is by coating another layer of soft material onto the surface of a hydrogel ^[1]. The coated material greatly suppresses the exchange of water molecules between the hydrogel and the external environment, thus making the hydrogel material stable. The coating method is simple and versatile and can be applied to a wide range of hydrogels ^[2]. The coated soft material can be diverse commonly used elastomers (e.g., polydimethylsiloxane Sylgard 184, Latex, VHB, and Ecoflex) ^[3], hydrophobic polymer ^[4], and hydrophilic substance ^[1]. The interface between the hydrogel and the coated soft material can be designed to be robust ^[3]. Moreover, the coated layer is thin and the coated soft material can also be highly stretchable, therefore, the coating method does not compromise the mechanical properties of hydrogels. For example, a polyacrylamide hydrogel with a 200- μm thick double-hydrophobic-coating exhibits strong resistance to both drying in air and swelling in aqueous environments ^[5]. The mass of coated hydrogel sample remains basically unchanged either placed in air or placed underwater for 7 days.

The second type of method to improve the water retention ability is introducing hygroscopic salt into the hydrogel. The dissociation of hygroscopic salts in water results in the formation of hydrated ions, decreasing the vapour pressure ^[6]. The presence of hygroscopic ions effectively strengthens the interactions between the water molecules within the hydrogel, preventing water evaporation ^[7]. The commonly adopted salts include lithium chloride, potassium acetate, magnesium chloride, and calcium chloride, among which lithium chloride is the most efficient one ^[8,9]. For example, a polyacrylamide hydrogel containing a high content of lithium chloride retains water at an extremely low relative humidity of 10% ^[10]. Moreover, the addition of hygroscopic ions transfers the hydrogel into a conductive material, making it a powerful candidate for applications such as wearable bioelectronic devices, ionic skins, and flexible electronics.

In the third method, the hydrogel is transferred to the organohydrogel by partially replacing water with hygroscopic solvents (e.g., glycerol, glycol, and sorbitol). The strong cooperative hydrogen bonding between the organic solvents and water molecules firmly anchored the water in the polymer network, and therefore endowed the organohydrogel with the properties of long-term stability ^[11]. The organohydrogel can be either directly polymerized in a glycerol–water binary solvent ^[12], or be completed by a facile solvent-exchange method ^[13]. A hydrogel becomes non-drying when some water is displaced with hygroscopic solvents. For example, an organohydrogel is fabricated by immersing a tough hydrogel in the glycerol solvent for three hours. The mass of the resulting glycerol-hydrogel was kept almost the same when it was placed at 20 °C and 50% relative humidity for 8 days ^[14].

The above-mentioned methods can be combined to achieve better stability of hydrogels. For instance, the combination of hygroscopic salt and hydrophobic coating eliminates dehydration ^[15].

We analyze the feasibility of combining the above methods with the strategy developed in this paper. In the current strategy, the hydrogel is loosely cross-linked and slightly dehydrated. The dehydrated hydrogel can be coated with elastomers and polymers just like the as-prepared hydrogels, therefore, the coating method is feasible.

Hygroscopic salts can be added into the precursor of hydrogels. Since the salt concentration increases with dehydration, less salt is needed in the precursor solution. The current strategy is possible to be extended to the organohydrogels. An as-prepared hydrogel is dehydrated first, then some of the remaining water is replaced with hygroscopic solvent. The resulting organohydrogel has a long chain structure and dehydration-induced entanglements, and it should have excellent anti-fracture and anti-fatigue ability, similar to the highly entangled hydrogel. In the current work, we focus on the hydrogel materials, and extending the current strategy to organohydrogels/organogels will be the focus of future work. The three methods to make a hydrogel more stable are summarized in Table. R1.

Table. R1 Analyzation of the combination of the proposed strategy and some methods to make a hydrogel more stable

Methods	Feasibility	Anti-drying	Anti-swelling in solution
Coating	Feasible	Yes	Yes
Adding hygroscopic salt	Feasible	Yes	No
Organohydrogel	Possible	Yes	No

[1] Gao, W., Chang, J., Li, X., Li, S., Zhou, Y., Hou, X., Long, L., Zhao, J., Yuan, X., 2023, A Quenched Double-Hydrophilic Coating for the Enhancement of Water Retention of Hydrogels, *Adv. Funct. Mater.*, 33, 2303306

[2] Li, X., Bian, F., Shi, J., Zhang, E., Kong, C., Ren, J., Wu, K., 2023, A novel environment-tolerant hydrogel via a combination effect of a polyurethane coating and hygroscopic salt for underwater monitoring, *J. Mater. Chem. C.*, 11, 5388-5401

[3] Yuk, H., Zhang, T., Parada, G., Liu, X., Zhao, X., 2016, Skin-inspired hydrogel-elastomer hybrids with robust interfaces and functional microstructures, *Nat. Commun.*, 7, 12028

[4] Zhu, T., Jiang, C., Wang, M., Zhu, C., Zhao, N., Xu, J., 2021, Skin - inspired double - hydrophobic - coating encapsulated hydrogels with enhanced water retention capacity, *Adv. Funct. Mater.*, 31, 2102433

[5] Mredha, M., Le, H., Cui, J., Jeon, I., 2020, Double - Hydrophobic - Coating through Quenching for Hydrogels with Strong Resistance to Both Drying and Swelling, *Adv. Sci.*, 7, 1903145

[6] Young, J., 1967, Humidity control in the laboratory using salt solutions—a review, *J. App. Chem.*, 17, 241-245

[7] Ge, W., Cao, S., Yang, Y., Rojas, O., Wang, X., 2021, Nanocellulose/LiCl systems enable conductive and stretchable electrolyte hydrogels with tolerance to dehydration and extreme cold conditions, *Chem. Eng. J.*, 408, 127306

[8] Sui, X., Guo, H., Cai, C., Li, Q., Wen, C., Zhang, X., Wang, X., Yang, J., Zhang, L., 2021, Ionic conductive hydrogels with long-lasting antifreezing, water retention and self-regeneration abilities, *Chem. Eng. J.*, 419, 129478

[9] Tang, L., Wu, S., Li, Y., Jiang, K., Xu, Y., Dai, B., Wang, W., Tang, J., Gong, L., 2023, A super-tough ionic conductive hydrogel with anti-freezing, water retention, and self-regenerated properties for self-powered flexible sensor, *Appl. Mater. Today.*, 32, 101820

[10] Bai, Y., Chen, B., Xiang, F., Zhou, J., Wang, H., Suo, Z., 2014, Transparent hydrogel with enhanced water retention capacity by introducing highly hydratable salt, *Appl Phys Lett*, 105, 151903

[11] Han, L., Liu, K., Wang, M., Wang, K., Fang, L., Chen, H., Zhou, J., Lu, X., 2018, Mussel - inspired adhesive and conductive hydrogel with long - lasting moisture and extreme temperature tolerance, *Adv. Funct. Mater.*, 28, 1704195

[12] Carvalho, F., Lopes, P., Carneiro, M., Serra, A., Coelho, J., Almeida, A., Tavakoli, M., 2020, Nondrying, sticky hydrogels for the next generation of high-resolution conformable bioelectronics, *ACS Appl. Electron. Mater.*, 2, 3390-3401

[13] Wu, J., Wu, Z., Xu, H., Wu, Q., Liu, C., Yang, B., Gui, X., Xie, X., Tao, K., Shen, Y., 2019, An intrinsically stretchable humidity sensor based on anti-drying, self-healing and transparent organohydrogels, *Mater. Horiz.*, 6, 595-603

[14] Chen, F., Zhou, D., Wang, J., Li, T., Zhou, X., Gan, T., Wang, S., Zhou, X., 2018, Rational fabrication of anti - freezing, non - drying tough organohydrogels by one-pot solvent displacement, *Angew. Chem. Int. Ed. Engl.*, 130, 6678-6681

[15] Floch, P., Yao, X., Liu, Q., Wang Z., Nian G., Sun, Y., Jia, L., Suo, Z., 2017, Wearable and washable conductors for active textiles, *ACS Appl. Mater. Interfaces*, 9, 25542-25552

4. In Figure 4b, the fracture toughness of the LC hydrogel exhibits a unique trend which first increase and then decrease with the reduction of water content. This is quite different from the trends observed for hydrogels with other cross-linking degrees. Could you provide further clarification on this?

Answer:

As the reviewer pointed out, the toughness of HC/MC hydrogels increases monotonously with the decreasing of water content; while for LC hydrogels, when the water content decreases, the toughness first increases then decreases, and the maximal toughness value corresponds to the water content of 70%.

The toughness of a hydrogel is positively related to its fracture stretch, energy dissipation ability, and the size of the energy dissipation area (i.e., the fractocohesive length). These properties are closely linked to the degree of entanglement. We calculated the ratio of the entangled modulus to the cross-linked modulus, α_{en} , serving as an indicator of the degree of entanglement.

For the HC/MC hydrogels, the ratio α_{en} increases from 0.4 to 2.6 and 0.7 to 7.0, respectively (see Supplementary Figure 10), as the water content decreases from 87% to 40%. The increase of entanglements improves both fracture stretch (see Fig.4c) and energy dissipation ability (see Fig.4f). The fractocohesive lengths exhibit no significant changes with variations in water content (see Supplementary Figure 7). In addition, the stress-stretch curve rises with the decrease of water content. As a result, the fracture toughness of the HC/MC hydrogels increases monotonously when the water content decreases.

For LC hydrogels with water contents of 87%, 70%, 55%, and 40%, the ratio α_{en} is calculated as 4.5, 38.9, 110.7, and 232.3, respectively. When the water content

decreases from 87% to 70%, despite a slight reduction in fractocohesive length (see Supplementary Figure 7), the fracture stretch, energy dissipation ability, and stress-stretch curve have all been improved. Therefore, the toughness first increases with decreasing water content. However, with a further reduction in water content, take the LC hydrogel with a water content of 55% as an example, the α_{en} increases rapidly to 110.7, indicating that the number density of entanglements becomes 110.7 times greater than that of the cross-linkers. Such dense entanglements take more time to remove, and some entanglements may not have sufficient time to unravel under a certain applied strain rate. Consequently, these entanglements that were not unraveled in time act as topologically physical cross-linkers. These additional physical cross-linkers reduce the stretchability of hydrogels obviously (see Fig. 4c). Furthermore, the entanglements that remain entangled during the deformation do not contribute significantly to energy dissipation, leading to a decrease in the energy dissipation ability (see Fig. 4f). What's worse, the fractocohesive length decreases sharply (see Supplementary Fig. 7). As a result, the toughness decreases when the water content further decreases from 70% to 40%.

To sum up, fracture toughness is highly related to the degree of entanglement. Within a suitable range, the toughness increases with the degree of entanglement (e.g., the HC/MC hydrogels). However, excessive DIEs, caused by heavy dehydration of LC hydrogels, are not conducive to achieving high toughness.

Some of the above discussions are exhibited in the first paragraph of the Discussion section.

5. Figure 4c presents an interesting result that the cracked sample with an initial water content of 87% exhibits a larger fracture stretch ratio. Is there any explanation for this phenomenon?

Answer:

We thank the reviewer for this question. This phenomenon is quite interesting. As shown in Fig. 4c, for the LC hydrogel with a water content of 87%, the fracture stretch of the precut samples is slightly larger than that of the uncut samples, i.e., $\lambda_c > \lambda_f$. Here, the sample dimensions are $50 \times 10 \times 2$ mm, with a precut crack length of 20 mm. The main reason for this unusual phenomenon is that the fractocohesive length of the material is larger than the length of the precut crack. As we mentioned in the manuscript, the fractocohesive length also describes a material-specific length that marks the transition from flaw-insensitive to flaw-sensitive fracture. Thus, the sample with a 20 mm-long crack here is flaw-insensitive, and the fracture stretch λ_c should be close to the fracture stretch λ_f . Experimentally, λ_c is slightly larger than λ_f , possibly caused by the reduction of lateral constraint due to the introduction of the precut crack.

This phenomenon can also be understood by comparing the following two equations: the calculation formula of toughness by pure shear test, $\Gamma = W(\lambda_c)H$, and the definition of the fractocohesive length, $\Gamma = W(\lambda_f)R_f$. Here, $W(\lambda)$ denotes the strain energy density of a deformed hydrogel with a stretch ratio of λ , H is the initial

height of the sample, and R_f stands for the fractocohesive length. We have $\Gamma = W(\lambda_c)H = W(\lambda_f)R_f$. For most hydrogels, the fractocohesive length is small ($R_f < H$), so the fracture stretch for precut sample is smaller than that of uncut samples ($\lambda_c < \lambda_f$). However, if the fractocohesive length exceeds the height of the sample, we will have $\lambda_c > \lambda_f$, as depicted in the unusual experimental observation in Fig. 4c. The above analysis can help us to understand why λ_c can be larger than λ_f .

6. During fatigue testing, it's assumed that the stress-strain curve reaches a steady state after 100 cycles. However, the maximum stress in Figure 5c seems to continue decreasing. If the load is sustained, would the stress-strain curve continue to decline? It might be beneficial to present cyclic curves over more cycles to provide a more comprehensive understanding of the material's fatigue behavior over an extended period.

Answer:

Thanks for this crucial suggestion. We agree that it is important to figure out the mechanical behaviors of the uncut hydrogels experimentally if more loading-unloading cycles are applied. In the previous manuscript, we showed the stress-stretch ratio (s-s) curves of 100 cycles, and here, we extend this to 1000 cycles. As illustrated in Fig. R9, the black line stands for the s-s curves of the initial 100 cycles, and the red line presents the s-s curves during the 101st cycle and the 1000th cycle. The loading after 100 cycles will cause further decay of the stress-stretch ratio curve, while the decrease of peak stress between the 101st and the 1000th cycle is measured as only 3.9% of the peak stress of the 1st cycle. Thus, the stress attenuation after 100 cycles is negligible.

We have included Fig. R9 in the Supplementary Information (Supplementary Fig. 9) to support the rationality for utilizing the 100th loading s-s curve to calculate the fatigue threshold.

Fig. R9. Nominal stress-number of cycles curve of an uncut LC hydrogel with a water content of 70% during 1000 cycles. The stress attenuation between the 101st cycle and the 1000th cycle is negligible, measuring 3.9% of the peak stress of the 1st cycle.

7. If a hydrogel is prepared directly with water content of 70%, bypassing the

dehydration process, would the result be same?

Answer:

A very good question. The answer to this question is No.

Adopting the current strategy, PAAm hydrogels with an initial water content of 87% and a current water content of 70% were prepared (Hydrogel *C* in Supplementary Table 2). Additionally, PAAm hydrogels with an initial water content of 70% were also prepared (Hydrogel *A* in Supplementary Table 2). Hydrogel *A* and hydrogel *C* share identical chemical components, i.e., a current water content of 70%, a cross-linker-to-monomer molar ratio of 5.0×10^{-5} , and an initiator-to-monomer molar ratio of 1.0×10^{-3} . Despite hydrogel *A* exhibiting a slightly higher modulus than hydrogel *C*, the latter is much tougher, with a fracture toughness measuring 21.8 times that of hydrogel *A*. In addition, its fatigue threshold is 4.1 times that of hydrogel *A*.

The difference of the mechanical properties arises from the processing and microstructures. Compared with polymerization-induced entanglements (PIEs), dehydration-induced entanglements (DIEs) have unique advantages in improving the toughness and fatigue threshold of hydrogels.

Some of the above analyses are exhibited in the Result section (The ninth line from the bottom of the first paragraph in Result - Outstanding overall mechanical properties).

REVIEWERS' COMMENTS

Reviewer #1 (Remarks to the Author):

The authors have resolved most of the comments raised by this Reviewer in the previous review. The manuscript has significantly improved, especially with the detailed experimental procedures provided for preparing uniformly dehydrated hydrogels and measuring their mechanical properties under cyclic testing in a humidity-controlled setup. This addition greatly enhances the clarity and comprehensibility of the work.

This Reviewer has one minor comment regarding the possibility of extending the DIE process to other types of polymeric hydrogels beyond polyacrylamide (PAAm). A simple demonstration of the DIE process on another hydrogel type would be very interesting and would further substantiate the versatility and broader applicability of the DIE method for enhancing the mechanical properties of various hydrogels.

Overall, the manuscript is well-written, and the innovative approach is effectively demonstrated. I look forward to seeing this additional example included to strengthen the manuscript further.

Reviewer #2 (Remarks to the Author):

The authors have well resolved the raised issues and revised the manuscript accordingly. The paper can be accepted now.

Response to reviewers' comments

We would like to thank the reviewers for their comments and thoughtful suggestions. The following are our responses to their comments:

Response to Reviewer 1's comments:

Reviewer 1 (Remarks to the Author):

The authors have resolved most of the comments raised by this Reviewer in the previous review. The manuscript has significantly improved, especially with the detailed experimental procedures provided for preparing uniformly dehydrated hydrogels and measuring their mechanical properties under cyclic testing in a humidity-controlled setup. This addition greatly enhances the clarity and comprehensibility of the work.

This Reviewer has one minor comment regarding the possibility of extending the DIE process to other types of polymeric hydrogels beyond polyacrylamide (PAAm). A simple demonstration of the DIE process on another hydrogel type would be very interesting and would further substantiate the versatility and broader applicability of the DIE method for enhancing the mechanical properties of various hydrogels.

Overall, the manuscript is well-written, and the innovative approach is effectively demonstrated. I look forward to seeing this additional example included to strengthen the manuscript further.

Answer:

We thank the reviewer for the positive comments and helpful suggestions.

In line with the reviewer's suggestion, we also paid attention to the generalizability and versatility of the proposed strategy. Our manuscript provides a theoretical analysis of this generalizability, as mentioned at the beginning of the 3rd paragraph of the Discussion section in our manuscript: "Entanglements exist in most hydrogels. When dehydrated, the polymer chains get closer to each other, forming dehydration-induced entanglements. In this sense, the proposed strategy is generally applicable to most hydrogel systems." Moreover, in fact, we have also tested another hydrogel, polyacrylic acid (PAAc), to verify the strategy's generalizability. The following analysis and discussion were provided in the 3rd paragraph of the Discussion part in our manuscript: "The current strategy was further applied to the polyacrylic acid (PAAc) hydrogel to verify its generality. Different from the PAAm hydrogel, the PAAc hydrogel features not only covalent cross-linking but also abundant physical interactions, such as hydrogen bonds, between the polymer chains. ... The fracture toughness of a regular PAAc hydrogel is 116.8 J m⁻². By the current strategy, the toughness of hydrogel C increased 57.8 times to 6753.7 J m⁻², and the modulus and fatigue threshold improved

by 3.0 times and 10.5 times in the meantime, respectively.". Therefore, we believe our manuscript adequately demonstrates the versatility of the developed strategy. In addition, we plan to expand this strategy to other polymer systems, such as organohydrogel, which will be the focus of our future work.

Response to Reviewer 2's comments:

Reviewer 2 (Remarks to the Author):

The authors have well resolved the raised issues and revised the manuscript accordingly. The paper can be accepted now.

Answer:

We are very grateful for the reviewer's positive comments.

REVIEWERS' COMMENTS

Reviewer #1 (Remarks to the Author):

The authors have resolved all issues raised previously. The manuscript can be accepted.
Congratulations!